# Psychomotor impairments and therapeutic implications revealed by a mutation associated with infantile Parkinsonism-Dystonia

Jenny I Aguilar[1,2], Mary Hongying Cheng[3], Josep Font[4], Alexandra C Schwartz[5], Kaitlyn Ledwitch[6,7], Amanda Duran[6,7], Samuel J Mabry[2], Andrea N Belovich[8], Yanqi Zhu[2], Angela M Carter[2], Lei Shi[9], Manju A Kurian[10], Cristina Fenollar-Ferrer[11], Jens Meiler[6,7,12], Renae Monique Ryan[4], Hassane S Mchaourab[5], Ivet Bahar[3], Heinrich JG Matthies[2†], Aurelio Galli[2,13†]*

[1]Department of Pharmacology, Vanderbilt University, Nashville, United States; [2]Department of Surgery, University of Alabama at Birmingham, Birmingham, United States; [3]Department of Computational and Systems Biology, School of Medicine, University of Pittsburgh, Pittsburgh, United States; [4]School of Medical Sciences, Faculty of Medicine and Health, University of Sydney, Sydney, Australia; [5]Department of Molecular Physiology & Biophysics, Vanderbilt University, Nashville, United States; [6]Center for Structural Biology, Vanderbilt University, Nashville, United States; [7]Department of Chemistry, Vanderbilt University, Nashville, United States; [8]Department of Biomedical Sciences, Idaho College of Osteopathic Medicine, Meridian, United States; [9]Computational Chemistry and Molecular Biophysics Section, NIDA, NIH, Baltimore, United States; [10]Molecular Neurosciences, Developmental Neurosciences, University College London (UCL), London, United Kingdom; [11]Laboratory of Molecular & Cellular Neurobiology, NIMH, NIH, Bethesda, United States; [12]Institute for Drug Discovery, Leipzig University Medical School, Leipzig, Germany; [13]Center for Inter-systemic Networks and Enteric Medical Advances, University of Alabama at Birmingham, Birmingham, United States

*For correspondence: agalli@uabmc.edu

†These authors contributed equally to this work

**Abstract** Parkinson disease (PD) is a progressive, neurodegenerative disorder affecting over 6.1 million people worldwide. Although the cause of PD remains unclear, studies of highly penetrant mutations identified in early-onset familial parkinsonism have contributed to our understanding of the molecular mechanisms underlying disease pathology. Dopamine (DA) transporter (DAT) deficiency syndrome (DTDS) is a distinct type of infantile parkinsonism-dystonia that shares key clinical features with PD, including motor deficits (progressive bradykinesia, tremor, hypomimia) and altered DA neurotransmission. Here, we define structural, functional, and behavioral consequences of a Cys substitution at R445 in human DAT (hDAT R445C), identified in a patient with DTDS. We found that this R445 substitution disrupts a phylogenetically conserved intracellular (IC) network of interactions that compromise the hDAT IC gate. This is demonstrated by both Rosetta molecular modeling and fine-grained simulations using hDAT R445C, as well as EPR analysis and X-ray crystallography of the bacterial homolog leucine transporter. Notably, the disruption of this IC network of interactions supported a channel-like intermediate of hDAT and compromised hDAT function. We demonstrate that *Drosophila melanogaster* expressing hDAT R445C show impaired hDAT activity, which is associated with DA dysfunction in isolated brains and

with abnormal behaviors monitored at high-speed time resolution. We show that hDAT R445C *Drosophila* exhibit motor deficits, lack of motor coordination (i.e. flight coordination) and phenotypic heterogeneity in these behaviors that is typically associated with DTDS and PD. These behaviors are linked with altered dopaminergic signaling stemming from loss of DA neurons and decreased DA availability. We rescued flight coordination with chloroquine, a lysosomal inhibitor that enhanced DAT expression in a heterologous expression system. Together, these studies shed some light on how a DTDS-linked DAT mutation underlies DA dysfunction and, possibly, clinical phenotypes shared by DTDS and PD.

## Introduction

Parkinson's disease (PD) is the second-most prevalent neurodegenerative disorder, affecting 2–3% of the global population over the age of 65 (*Chai and Lim, 2013*). Although a vast majority of PD cases occur idiopathically and affect people over the age of 50 (late-onset), a subset of genetic mutations is associated with early-onset PD (*Lill, 2016*). Investigations of these highly penetrant, inherited forms of PD have provided tremendous insights into specific molecular pathways that underlie neurodegeneration and motor deficits (*Trinh and Farrer, 2013*).

Mutations in the human dopamine (DA) transporter (hDAT) gene (*SLC6A3*) have been linked to a distinct type of infantile parkinsonism-dystonia, referred to as DA transporter deficiency syndrome (DTDS) (*Kurian et al., 2011*; *Kurian et al., 2009*; *Ng et al., 2014*). Few patients diagnosed with DTDS survive to adulthood, with a majority of patients dying in childhood or adolescence (*Kurian et al., 2011*; *Kurian et al., 2009*; *Ng et al., 2014*). Common to DTDS-linked DAT variants is a multifaceted loss of DAT function, which includes impaired transporter activity and decreased expression (*Asjad et al., 2017*; *Beerepoot et al., 2016*; *Kurian et al., 2011*; *Kurian et al., 2009*; *Ng et al., 2014*). However, the structural and functional underpinnings of these impairments, how they translate to specific behaviors, and whether they can be pharmacologically targeted remain mostly uncovered.

DTDS is a complex movement disorder typically characterized by initial infantile hyperkinesia (dyskinesia/dystonia) that progresses to a parkinsonian movement disorder (bradykinesia/tremor) (*Kurian et al., 2011*; *Ng et al., 2014*). Other characteristic clinical features include elevated levels of the DA metabolite homovanillic acid (HVA) in the cerebrospinal fluid and loss of DAT activity in the basal ganglia, as measured by single-photon emission tomography of DAT (i.e. DaTSCAN) (*Kurian et al., 2011*; *Kurian et al., 2009*; *Ng et al., 2014*). Increased levels of HVA might reflect increased DA turnover promoted by higher extracellular DA levels. This increase in DA levels could stem from decreased DA clearance mediated either by loss of DAT activity and/or expression. Other forms of early-onset parkinsonism are associated with impaired DAT function (*Hansen et al., 2014*). This includes a patient with early-onset parkinsonism and attention deficit hyperactivity disorder (ADHD) carrying compound heterozygous missense mutations in *SLC6A3* that give rise to I321F and D421N substitutions in the DAT protein (*Borre et al., 2014*; *Hansen et al., 2014*). To date, the mechanism through which altered DAT function underlies parkinsonian phenotypes remains unclear.

The DAT is a presynaptic membrane protein that spatially and temporally regulates DA neurotransmission by mediating the reuptake of DA from the synapse following vesicular release. Among other roles, DA regulates cognition, emotion, motor activity, and motivation (*Björklund and Dunnett, 2007*; *Giros and Caron, 1993*; *Palmiter, 2008*). Altered DA neurotransmission has been implicated in several neuropsychiatric and neurological disorders, including ADHD, Autism Spectrum Disorder (ASD) and PD (*Bowton et al., 2010*; *Bowton et al., 2014*; *Cartier et al., 2015*; *Chai and Lim, 2013*; *NIH ARRA Autism Sequencing Consortium et al., 2013*; *Meisenzahl et al., 2007*; *Russo and Nestler, 2013*; *Swanson et al., 2007*). Structural and molecular dynamic (MD) studies suggest that DA transport occurs via an alternating access model, wherein the transporter alternates between various 'outward-facing' and 'inward-facing' conformations (*Forrest et al., 2008*; *Kazmier et al., 2014*; *Krishnamurthy and Gouaux, 2012*). hDAT can also form an aqueous pore (channel-like mode) (*Bowton et al., 2010*; *Bowton et al., 2014*; *Kahlig et al., 2005*). We have shown that the frequency of the hDAT channel-like mode is enhanced by both pharmacological targeting and disease-associated variants (*Bowton et al., 2010*; *Bowton et al., 2014*; *Kahlig et al.,*

*2005*). Key to this alternating mechanism is a network of interactions occurring at the extracellular (EC) and intracellular (IC) transporter face, termed EC and IC gates, respectively.

Recent work identified compound heterozygous missense mutations in the *SLC6A3* gene in a patient who presented with classical DTDS: a mutation in one allele resulted in a R445C substitution and a mutation in the second allele resulted in a R85L substitution (*Ng et al., 2014*). Either mutation, when studied individually, has devastating effects on hDAT activity and expression (*Ng et al., 2014*). In this study, we aimed to understand, mechanistically and structurally, how R445C disrupts transport function, the behavioral consequence of this disruption, as well as whether we could rescue transport function and behaviors with pharmacotherapy. Of note, DAT has been shown to form both dimers as well as tetramers at the plasma membrane (*Hastrup et al., 2003*). Therefore, expression of both mutations in our experimental preparations would generate several combinations of hDAT oligomers, preventing the association of specific hDAT impairments and behavioral phenotypes to a specific mutation. Thus, we focused this study on the R445C mutation.

R445 is located close to the cytoplasmic end of TM9, facing the IC vestibule and is part of a conserved IC interaction network that comprises the IC gate (*Khelashvili et al., 2015a*; *Kniazeff et al., 2008*; *Razavi et al., 2018*; *Reith et al., 2018*; *Shan et al., 2011*). This network is thought to coordinate conformational rearrangements in DAT throughout the transport cycle (*Kniazeff et al., 2008*; *Shan et al., 2011*). Specifically, the R445-E428 salt bridge is predicted to stabilize the transition of hDAT to an inward-occluded conformation (*Khelashvili et al., 2015a*; *Penmatsa et al., 2013*; *Reith et al., 2018*). Previous studies showed that substitutions at R445 impair DAT function (*Asjad et al., 2017*; *Beerepoot et al., 2016*; *Ng et al., 2014*; *Reith et al., 2018*). However, how and whether R445C impacts the structure and the dynamics of the IC gate remains unclear. Importantly, how the R445C substitution contributes to DA dysfunction in disease and more specifically, DTDS etiology, is largely unknown.

Here, we undertake a close examination of the structural and functional consequences of the R445C substitution in hDAT. We integrate molecular insights from X-ray crystallography, electron paramagnetic resonance (EPR), molecular modeling, and molecular dynamic (MD) simulations to determine, mechanistically, how R445C underlies dysfunction of the DAT IC gate. Furthermore, we adopt *Drosophila melanogaster* as an animal model to examine whether and how this hDAT variant supports brain DA dysfunction, loss of DA neurons and behavioral phenotypes characterized by DTDS. Finally, we assess a pharmacological agent for its ability to rescue behavioral deficits in *Drosophila* expressing hDAT R445C. Together, this work provides insight into the structural mechanisms underlying DAT dysfunction and the impact of DAT dysfunction on specific behaviors, as well as on the molecular mechanisms that underlie DTDS and more broadly, PD pathology.

## Results

### hDAT R445C compromises movement vigor in *Drosophila*

*Drosophila melanogaster* have provided unique and critical insights on the pathogenic mechanisms underlying PD (*Feany and Bender, 2000*; *Xiong and Yu, 2018*). *Drosophila* PD models consistently recapitulate essential PD phenotypes, including neurodegeneration as well as motor and non-motor behavioral deficits (*Nagoshi, 2018*). In addition, mechanisms that mediate DA neurotransmission and signaling observed in other phyla are largely conserved in *Drosophila* (*Yamamoto and Seto, 2014*). As observed in mammals, *Drosophila* exhibit increased arousal and hyperactivity, among other stereotypies, when DAT function is altered (*Kume et al., 2005*; *McClung and Hirsh, 1998*).

In order to understand whether certain DAT dysfunctions are associated with specific phenotypes in *Drosophila*, we assessed whether the R445C missense mutation in the DAT promoted behaviors associated with common DTDS phenotypes. We adopted the Gal4/UAS system to express hDAT WT or hDAT R445C specifically in DA neurons of flies homozygous for the *Drosophila* DAT null allele (*DAT^{fmn}*) (*Campbell et al., 2019*; *Cartier et al., 2015*; *NIH ARRA Autism Sequencing Consortium et al., 2013*). This system has two parts: the Gal4 gene, encoding the yeast transcription activator protein Gal4, and the upstream activation sequence (UAS), a minimal promoter region to which Gal4 specifically binds to activate the transcription of the gene of interest (in this study, hDAT). We developed flies where Gal4 expression is driven by the tyrosine hydroxylase (TH) promoter (TH-GAL4), driving the expression of Gal4 specifically in DA neurons (in flies, octopamine, the *Drosophila* analog

of norepinephrine, does not require TH for synthesis *Blenau and Baumann, 2001*). The gene of interest is inserted into an attB donor plasmid with a UAS site (*Bischof et al., 2007*). This approach allows for irreversible integration of the gene of interest into the identical genomic locus *via* an integrase (phiC31) through the integrated phage *att*achment site, *attP* (the recipient site in the Drosophila genome). This leads to expression of comparable levels of mRNA for transgenes (e.g. hDAT). These transgenic organisms are generated with no need for mapping of the insertion site (*Bischof et al., 2007*).

We first determined that utilizing the Gal4/UAS system to express hDAT WT specifically in DA neurons of *DAT^{fmn}* flies does not alter DA-associated phenotypes compared to wild type animals expressing *Drosophila* DAT (dDAT). To do this, we investigated whether hDAT WT, in *Drosophila* brains, could support the reverse transport (efflux) of DA evoked by amphetamine (AMPH) as observed for dDAT. The psychostimulant AMPH evokes DA efflux mediated by the DAT (*Robertson et al., 2009*). To measure DA efflux by amperometry, we guided a carbon fiber electrode into the *Drosophila* brain juxtaposed to the mCherry-tagged posterior inferior lateral protocerebrum (PPL1) cluster of DA neurons (see below for details) (*Shekar et al., 2017*) In *Figure 1—figure supplement 1*, representative traces display current measurements of AMPH (20 μM)-induced DA efflux from this population of neurons from *DAT^{fmn}*, dDAT, and hDAT WT fly brains (A-top). Quantitation of correspondent peak currents demonstrate comparable efflux for dDAT and hDAT (A-bottom). Cocaine (20 μM), a DAT blocker, inhibited the ability of AMPH to cause DA efflux in hDAT WT brains (*Figure 1—figure supplement 1A*, top). Further, we determined that uptake of [$^3$H]DA in hDAT WT *Drosophila* brains was not significantly different from that measured in dDAT brains (*Figure 1—figure supplement 1B*). The absence of uptake in the *DAT^{fmn}* fly brains shows the dependence of DA uptake on the DAT.

In *Drosophila*, locomotion is regulated by DA neurotransmission as well as DAT function (*Campbell et al., 2019*; *Cartier et al., 2015*; *Hamilton et al., 2014*; *NIH ARRA Autism Sequencing Consortium et al., 2013*; *Pizzo et al., 2014*). We have previously demonstrated that AMPH causes changes in locomotion, a behavior that depends on DAT function/expression (*Cartier et al., 2015*; *Hamilton et al., 2014*). Adult *Drosophila* males were fed a sucrose solution (5 mM) containing either AMPH (1 mM) or vehicle (CTR). Locomotion was measured by beam crossing detection over a 60 min period. In *Drosophila* expressing dDAT, AMPH significantly stimulates locomotion (*Figure 1—figure supplement 1C*). Remarkably, in *DAT^{fmn}* flies, AMPH did not increase locomotion. These data demonstrate that in adult *Drosophila*, functional DAT is required for AMPH-induced locomotion. To support this animal model for studying how changes in hDAT function affects behaviors, we rescued AMPH-induced locomotion in the *DAT^{fmn}* flies by expressing hDAT selectively in DA neurons using the Gal4/UAS system (*Figure 1—figure supplement 1C*). These data strongly support *Drosophila* as a model system to test the multiple functions of hDAT in vivo.

We tested flies for spontaneous locomotor activity and 'anxiety'-related behaviors, such as time spent in or near the center of an enclosure during an open-field test (i.e. center time). Illustrated are representative trajectories of adult hDAT WT flies (*Figure 1A*, *black trace*) and hDAT R445C flies (*Figure 1A*, *blue trace*) assayed in an open-field test for 5 min. We observed no differences in center time in hDAT R445C flies with respect to hDAT WT flies (*Figure 1B*; hDAT WT: 0.016 ± 0.003 (t/t$_{total}$); hDAT R445C: 0.024 ± 0.006 (t/t$_{total}$); p>0.05). We did observe a significant reduction in spontaneous locomotor activity in hDAT R445C (59.7 ± 6.1 cm) compared with hDAT WT flies (80.1 ± 4.2 cm; p=0.008) (*Figure 1C*). Given that parkinsonian locomotor deficits can be characterized by hypokinesia (inability to initiate movement) and bradykinesia (slowed movement), we dissected the specific locomotor deficits observed in hDAT R445C flies. We determined the frequency with which specific velocities were explored throughout the test period (*Figure 1D*). We defined 'initiating movement' as velocity = 0.74–0.94 mm/s and 'fast movement' as velocity = 5.3–10.0 mm/s and determined their frequency per genotype. hDAT R445C flies spent 5.0 ± 0.3% of the testing period initiating movement compared with 5.2 ± 0.4% for hDAT WT flies, suggesting hDAT R445C flies did not have difficulty performing this task (p>0.05; *Figure 1E*). In contrast, hDAT R445C flies displayed significantly decreased movement vigor, in fast movement for only 9.8 ± 1.4% of the testing period compared with 14.5 ± 1.1% for hDAT WT flies (p=0.0098; *Figure 1F*). Together, these data suggest that motor deficits in hDAT R445C flies are primarily characterized by deficits in movement vigor.

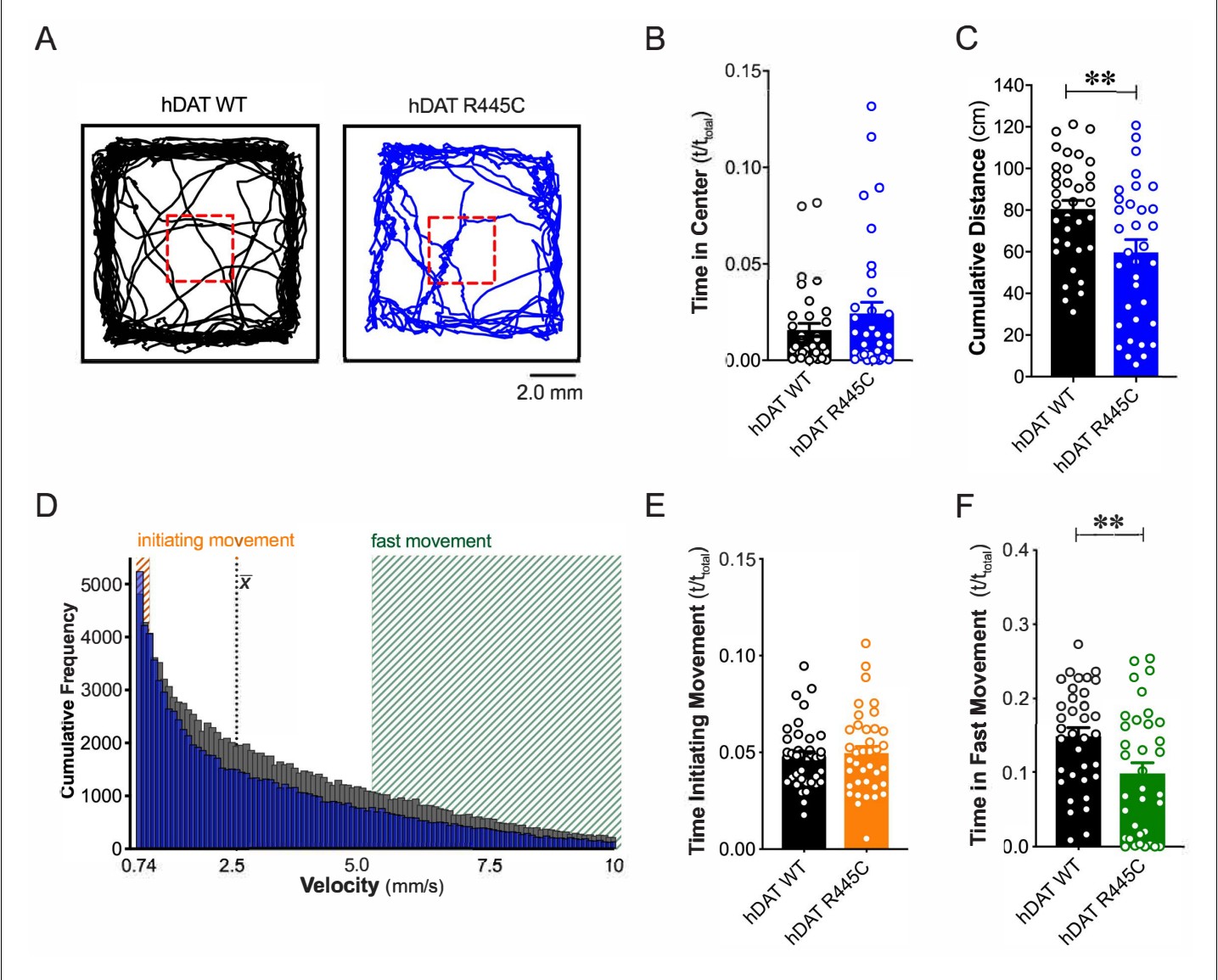

**Figure 1.** R445C variant disrupts locomotive behaviors in *Drosophila*. (A) hDAT WT or hDAT R445C was selectively expressed in DA neurons in a dDAT KO (*fmn*) background. Representative trajectories of hDAT WT (*black*) and hDAT R445C (*blue*) flies in an open-field test during a 5 min test period. 3 × 3 mm square (*red dashed lines*) delineates the center space. (B) hDAT WT and hDAT R445C flies spent comparable time in the center space (p>0.05; n = 35). (C) hDAT R445C flies traveled significantly less relative to hDAT WT flies (p=0.006; n = 35). (D) Histogram represents instantaneous velocities ranging from 0.74 to 10.0 mm/s (bin width = 0.094 mm/s; see Materials and methods) and corresponding frequencies (number of times) for hDAT WT (*gray bars*) and hDAT R445C (*blue bars*) flies. Initiating movement velocities (0.74–0.94 mm/s, *orange shaded*), fast movement velocities (5.3–10.0 mm/s, *green shaded*), average velocity ($\bar{x}$) are highlighted. (E) hDAT R445C flies spent a comparable amount of time initiating movement relative to hDAT WT flies (p>0.05; n = 35). (F) hDAT WT flies spent significantly more time in fast movement compared with hDAT R445C flies (p=0.001; n = 35). Data represent mean ± SEM. Welch's t-test: (B); Mann-Whitney test (C) and (E - F).

The online version of this article includes the following figure supplement(s) for figure 1:

**Figure supplement 1.** AMPH-induced DA efflux and behaviors in hDAT WT flies.

## hDAT R445C impairs selective coordinated movements

Patients with early-onset as well as sporadic PD often present impairments in coordination (*van den Berg et al., 2000*). To understand further the contribution of the DAT to coordinated motor behaviors, we analyzed a quintessential fly behavior: flight. Various monoamines, including DA, modulate insect flight (*Sadaf et al., 2015*). Inhibition of specific TH-positive DA neurons has been found to compromise flight, including impaired wing coordination and kinematics (*Sadaf et al., 2015*).

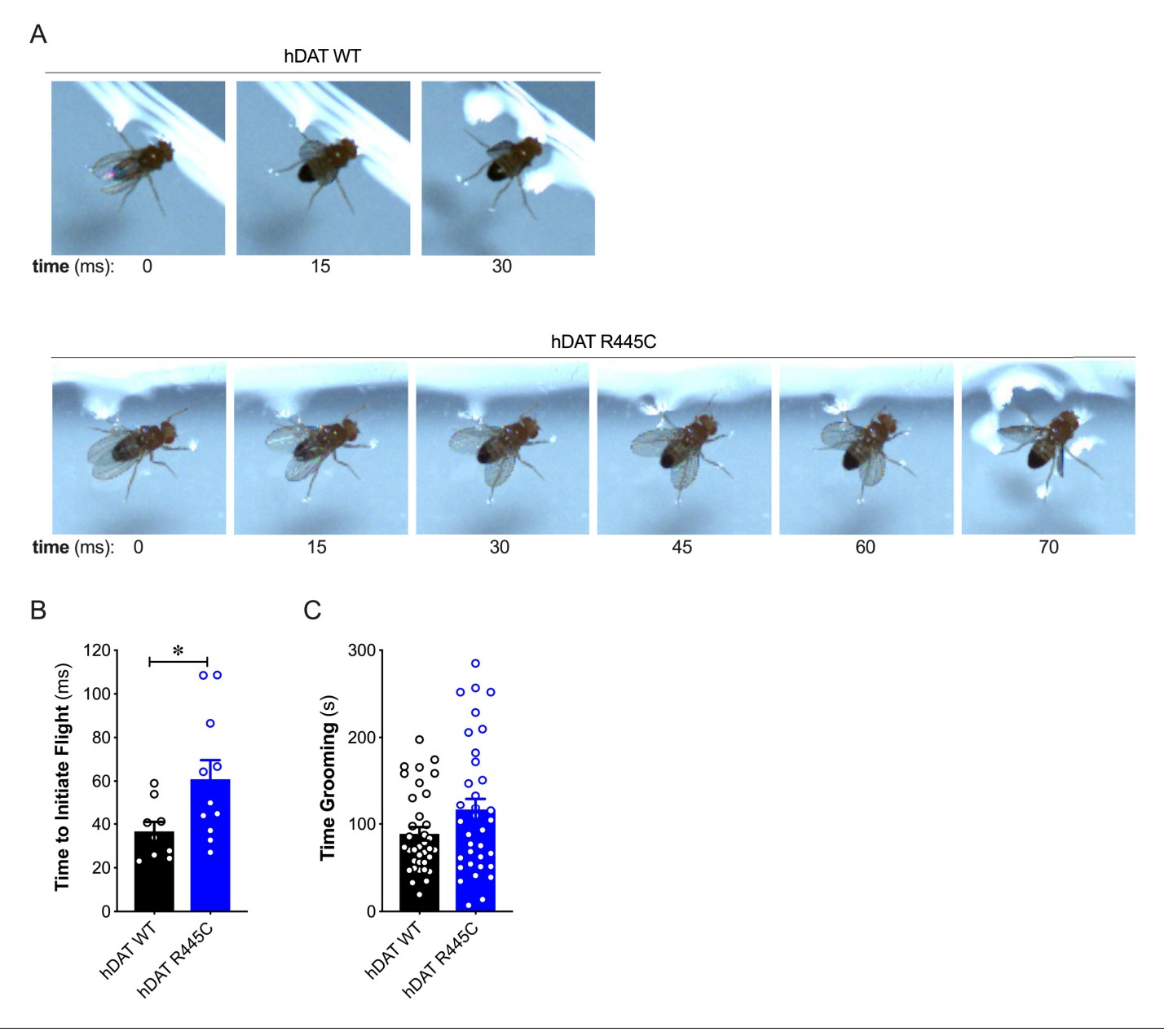

**Figure 2.** R445C variant selectively impairs coordinated motor behaviors, such as take-off, but not repetitive motor behaviors, such as grooming. (A) Representative single frames of *Drosophila* hDAT WT (*top*) and hDAT R445C (*bottom*) during various phases of coordinated take-off (video recorded at 2000 fps). (B) Flight initiation (take-off) was quantified from the initial phase of wing elevation (t = 0) to the second phase of simultaneous wing depression and leg extension. Flight initiation was significantly delayed in hDAT R445C flies relative to hDAT WT (p=0.03; n = 10–11). (C) hDAT R445C flies spent comparable time grooming compared with hDAT WT flies (p>0.05; n = 35). Data represent mean ± SEM. Welch's t-test: (B); Mann-Whitney test (C).

Initiating voluntary flight (take-off) consists of an initial phase of wing elevation, followed by a second phase of simultaneous left- and right-wing depression and leg extension (*Zabalax et al., 2008*). Using a high-speed camera (2000 fps), we quantified the time that elapsed between the initiation of wing elevation (t = 0) and final take-off from a water surface (*Figure 2A* and *Videos 1–2*). We found that flight initiation was significantly compromised in hDAT R445C flies as the corresponding duration of take-off was 60.9 ± 8.7 ms compared with 36.6 ± 4.4 ms for hDAT WT flies (p=0.03). To determine whether impairments in coordination were consistent across multiple modalities, we assessed grooming. In *Drosophila,* this stereotyped, coordinated movement of the forelegs and

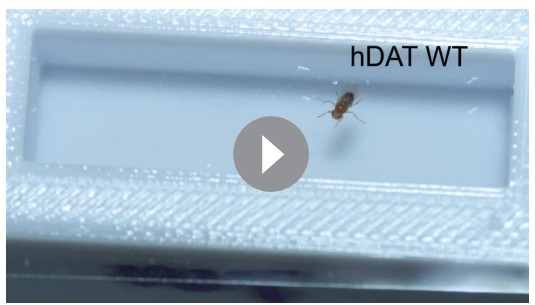

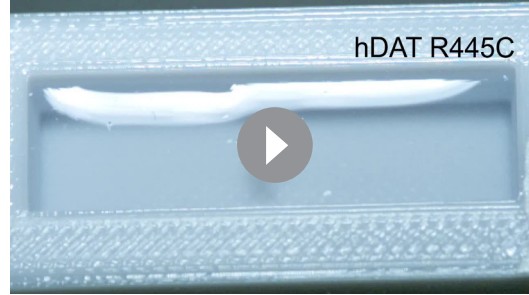

**Video 1.** hDAT WT flies in flight. Representative video of coordinated flight take-off of *Drosophila* hDAT WT. Video display is slowed down from the 'start' to 'stop' time of take-off to ease visualization.
https://elifesciences.org/articles/68039#video1

**Video 2.** hDAT R445C flies in flight. Representative video of coordinated flight take-off of *Drosophila* hDAT R445C. Video display is slowed down from the 'start' to 'stop' time of take-off to ease visualization.
https://elifesciences.org/articles/68039#video2

hindlegs is prompted by a mechanical or microbial stimulus and is modulated by dopaminergic neurotransmission (*Pitmon et al., 2016*). Interestingly, grooming was not significantly impaired in hDAT R445C flies (116.7 ± 12.9 s) relative to hDAT WT flies (88.8 ± 7.7 s; p>0.05) (*Figure 2C*). These data suggest that only specific coordinated movements are impaired in hDAT R445C flies.

## hDAT R445C flies display DA deficiency

DA dysregulation, specifically the loss of DA signaling, drastically alters the timing, velocity and fluidity with which movement is executed (*Panigrahi et al., 2015*; *Turner and Desmurget, 2010*). We thus sought to determine whether impairments in movement and coordination were driven by altered DA dynamics. We first measured DA content in whole brains of hDAT WT and hDAT R445C flies. DA content was significantly reduced by 16.9 ± 3.2% in hDAT R445C (21.4 ± 0.8 ng/mg) relative to hDAT WT brains (25.8 ± 1.0 ng/mg) (p=0.02) (*Figure 3A*, left). We also measured serotonin (5-HT) content, as serotonergic dysfunction has also been associated with the development of motor and non-motor symptoms in PD (*Politis and Niccolini, 2015*). We found that 5-HT content was comparable in hDAT WT (67.0 ± 1.8 ng/mg) and hDAT R445C (60.7 ± 2.1 ng/mg; p>0.05) brains (*Figure 3A*, right).

Various *Drosophila* PD models have shown selective neurodegeneration of protocerebral posterior lateral 1 (PPL1) DA neurons (*Barone et al., 2011*; *Cackovic et al., 2018*; *Trinh et al., 2008*; *Whitworth et al., 2005*). These clusters of neurons, which innervate the mushroom and fan-shaped bodies, are implicated in regulating motivated behaviors as well as reward learning and reinforcement. Thus, they exhibit parallel functions compared to DA projections from the substantia nigra to the striatum in mammals (*Aso et al., 2012*; *Berry et al., 2012*; *Claridge-Chang et al., 2009*; *Kirkhart and Scott, 2015*; *Riemensperger et al., 2011*). We assessed the number of TH-positive PPL1 neurons in hDAT WT and hDAT R445C brains (*Figure 3B*, left). We found TH-positive PPL1 neurons to be significantly reduced in hDAT R445C flies (9.1 ± 0.4) relative to hDAT WT controls (11.5 ± 0.2; p<0.0001) (*Figure 3B*, right). These data demonstrated that specific motor deficits are associated with DA deficiency in hDAT R445C flies.

To determine the effects of R445C on DAT function, we examined reverse transport (efflux) of DA evoked by amphetamine (AMPH) in isolated *Drosophila* brains. The psychostimulant AMPH evokes DA efflux mediated by the DAT. To measure DA efflux, we utilized amperometry in isolated *Drosophila* brains. We guided a carbon fiber electrode into the brain, juxtaposed to the mCherry-tagged PPL1 DA neurons (*Shekar et al., 2017*; *Figure 3C*, left, red box). The representative traces displayed are amperometric current measurements of DA efflux from this population of neurons in hDAT WT and hDAT R445C brains (*Figure 3C*, middle). Given the DA deficiency in hDAT R445C brains, it was not surprising that AMPH-induced DA efflux was significantly reduced in hDAT R445C (0.76 ± 0.14 pA) compared with hDAT WT (1.74 ± 0.37 pA; p=0.04) brains. Nonetheless, these brains were capable of DA efflux, suggesting that hDAT R445C can support, at least in part, the reverse transport of DA.

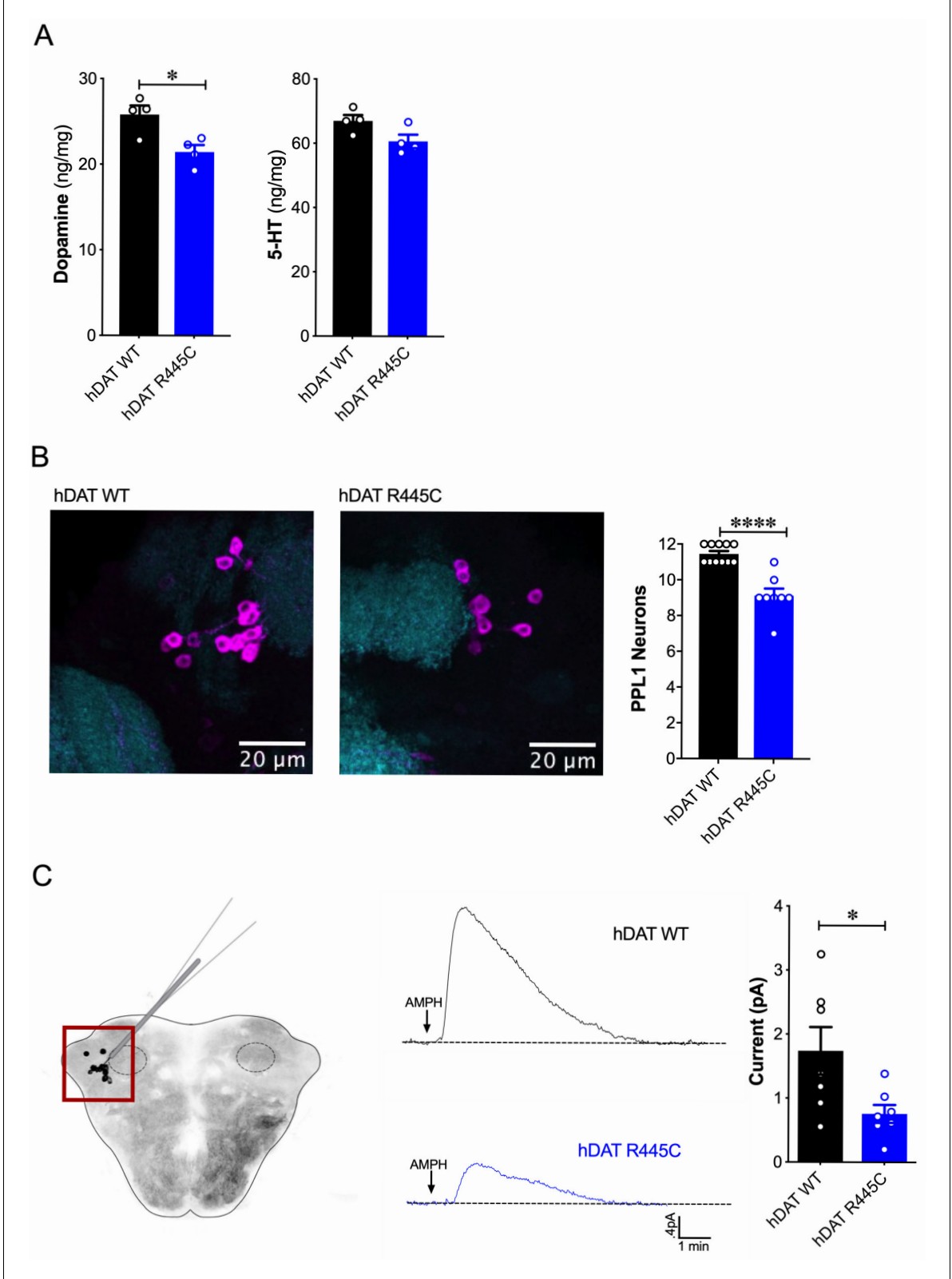

**Figure 3.** Reduced brain DA content and AMPH-induced DA efflux in hDAT R445C flies. (**A**) Tissue concentration of DA (*left*) and 5-HT (*right*) measured by HPLC (n = 4, 20 brains per measurement). DA content was significantly decreased in hDAT R445C relative to hDAT WT brains (p=0.01). 5-HT content in hDAT R445C was comparable to hDAT WT (p>0.05). (**B**) Confocal z-stack (5 μm) of hDAT WT (*left*) and hDAT R445C (*center*) brains co-stained with anti-TH (*magenta*) and anti-n82 (*cyan*) zoomed-in on PPL1 neurons. Quantitation of TH-positive PPL1 neurons showed a significant reduction of these

*Figure 3 continued on next page*

*Figure 3 continued*

neurons in hDAT R445C brains relative to hDAT WT (p<0.0001; n = 8–11; *right*) (**C**) Diagram illustrates amperometric studies in *Drosophila* brains in which a carbon fiber electrode records currents from TH-positive PPL1 DA neuronal region (*red box*) in response to AMPH application (20 µM; *left*). AMPH-induced (arrow) amperometric currents in hDAT WT (*black trace*) and hDAT R445C (*blue trace*) brains. Quantitation of peak currents showed a significant decrease in DA efflux measured in hDAT R445C relative to hDAT WT (p=0.04; n = 7; *right*). Data represent mean ± SEM. Student's t-test (**A**)-(**B**); Welch's t-test: (**C**).

## Substitutions in LeuT, at the site homologous to R445 in hDAT, disrupt IC network interactions

LeuT, the bacterial homolog of hDAT, has provided key insights that have improved our understanding of $Na^+$/substrate-coupled transport in the neurotransmitter sodium symporter (NSS) family (*Beuming et al., 2006*; *Yamashita et al., 2005*). Integrating data from LeuT crystal structures, electron paramagnetic resonance (EPR), single-molecule fluorescence energy transfer (sm-FRET) and MD simulations has defined the alternating access mechanism used by the NSS family to transport substrate. Common to these models is the transition from outward-facing open (OF) to inward-facing open (IF) states through the opening and closing of the EC and IC gates, respectively (*Claxton et al., 2010*; *Kazmier et al., 2014*). Here, we use a combination of Rosetta modeling, X-ray crystallography and EPR spectroscopy to determine the consequence of hDAT mutations at R445 on conformational changes in LeuT.

Previous studies of LeuT conformational dynamics have shown that the network of interactions between the N-terminus (residues R5, E6, W8), TM6/IL3 (Y265, Y268), TM8 (D369), and TM9 (R375) are key to occluding the IC vestibule in the outward-facing occluded (OO) state (*Cheng and Bahar, 2014*). In particular, salt bridges R5-D369 and E6-R375 stabilize the N-terminus in the OO state, as illustrated in *Figure 4A* (**left**). The residues participating in this network are highly conserved across the NSS family, and are thus, likely critical to transport. First, we determined the effects of substitutions at the LeuT residue corresponding to R445 of hDAT, R375 in LeuT: LeuT R375A, LeuT R375D, and LeuT R375C. We constructed molecular models of LeuT R375A and R375D (*Figure 4A*) using Rosetta to determine potential changes in these interactions and in the thermodynamic stability (ΔΔG) of these variants relative to WT. We found that both neutralizing and acidic substitutions at R375 likely promote the dissociation of the E6-R375 salt bridge (closest atom-atom distances: WT = 2.1 Å; R375A = 5.4 Å; R375D = 4.2 Å), weaken the interaction of R375 and I184 (closest atom-atom distances: WT = 2.5 Å; R375A = 6.0 Å; R375D = 4.7 Å), and decrease the thermodynamic stability of LeuT (Rosetta scores: R375A = +4.4 REU; R375D = +5.6 REU relative to WT) (*Figure 4A*, *Figure 4—figure supplement 1A–B*). Other interactions were largely preserved, including R5-D369 and E6-I187 interactions (*Figure 4A*). One difference between these models was that K189 moved towards E6 in LeuT R375A, but away from E6 in LeuT R375D (*Figure 4—figure supplement 1A*). Together, these models predicted that both acidic and neutral mutations at the LeuT counterpart (R375) of hDAT R445 disrupt the interactions near the IC vestibule, partially affecting the IC gate, but maintaining other IC network interactions. We also generated a model for LeuT R375C, and found that K189 also moves away from E6. We conclude that a cysteine mutation at R375 more closely resembles an acidic substitution (compare *Figure 4A* and *Figure 4—figure supplement 1A, C*). However, it has to be noted that cysteine residues exist at an equal ratio of protonated (neutral) to deprotonated (acidic) states at physiological pH. The root-mean-square deviation (RMSD) was calculated to show the correlation between the energy-optimized models and the experimental model (*Figure 4—figure supplement 1D*).

To define further the structural consequences of R375 substitutions, we determined the X-ray crystal structures of LeuT WT, LeuT R375A, and LeuT R375D solved in the L-Ala and $Na^+$ bound OO conformation to a resolution of 2.1 Å for WT and R375A, and 2.6 Å for R375D (*Figure 4B*, detailed in *Figure 4—figure supplement 2*). Unfortunately, the expression of the R375C mutant was low and protein yield was insufficient for crystallography. Structures were aligned with a previous structure of LeuT WT in an OO conformation (PDB ID: 3F3E) with an RMSD of 0.134, 0.146, and 0.236 for LeuT WT, R375A, and R375D, respectively. In all structures (*superimposed*), L-Ala, Na1, and Na2 (*purple spheres*) could be modeled into their respective binding sites (*Figure 4B*, left). These crystal

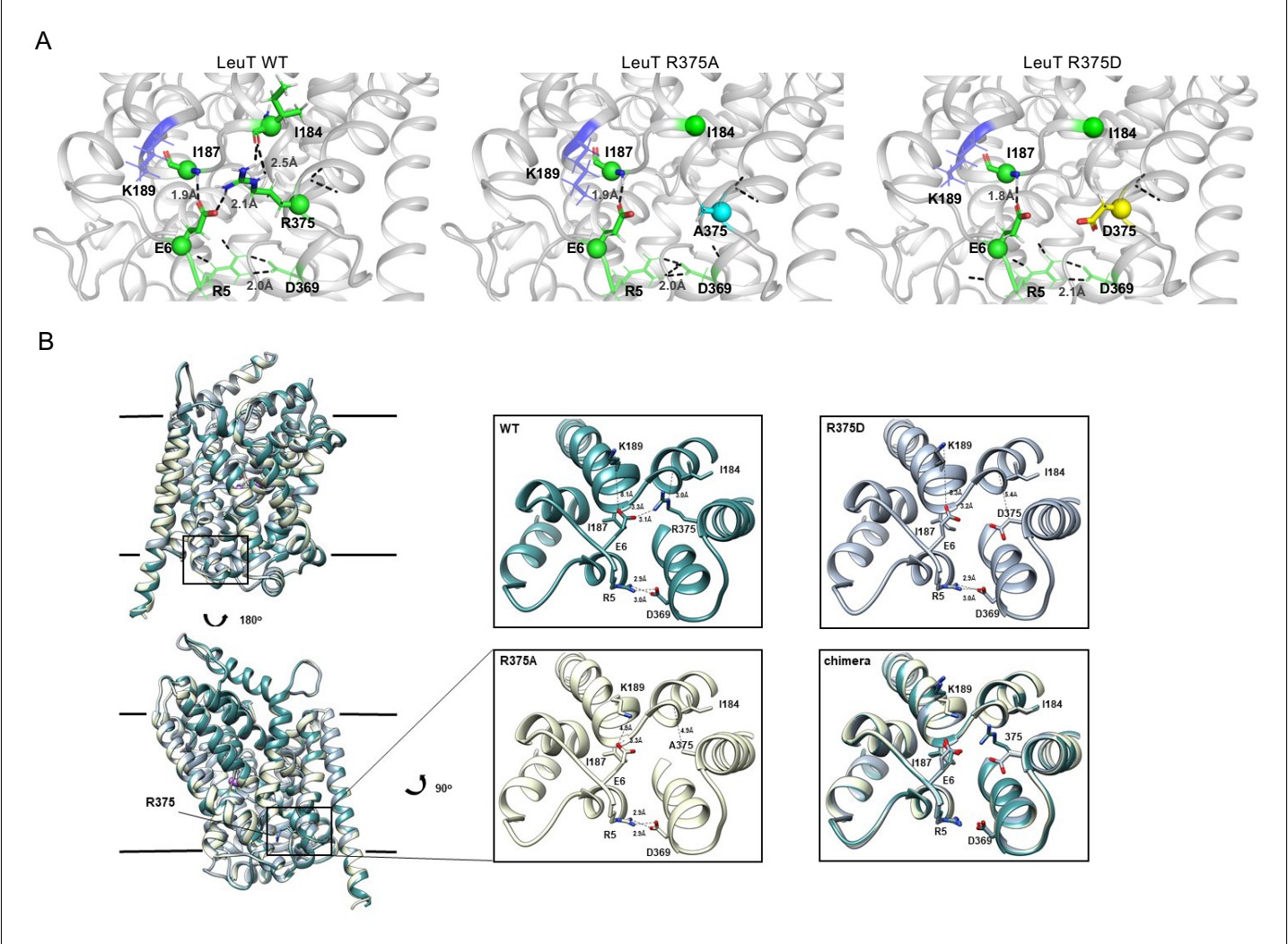

**Figure 4.** Representative Rosetta models and crystallographic structures of LeuT revealed weakening of E6-R375 salt bridge promoted by substitutions at R375 (corresponding to R445 in hDAT). (A) Models derived, using Rosetta, of LeuT WT (*left*), LeuT R375A (*middle*) and LeuT R375D (*right*) where protein backbones are represented as cartoons and residues E6, I184, I187, R5, and D369 are represented as green spheres and sticks. K189 is colored in blue throughout each model. R375 is colored in green (*left*). A375 is colored in cyan (*middle*). D375 is colored in yellow (*right*). All corresponding polar contacts between side chain or backbone atoms in each model are represented as dashed lines in black. R375 substitution to either Ala or Asp disrupted E6-R375 salt bridge. (B) Crystal structures of LeuT WT (*green*), LeuT R375A (*cream*) and LeuT R375D (grey) are superimposed. Box indicates area of zoomed-in view of TM1-TM8 IC region for LeuT WT (top *left*), LeuT R375D (top *right*), LeuT R375A (bottom left) and overlay of three structures (*bottom right*). Distances between residues are shown in dashed lines.

The online version of this article includes the following figure supplement(s) for figure 4:

**Figure supplement 1.** Related to *Figure 4A*.
**Figure supplement 2.** Related to *Figure 4B*.

structures showed that R375A and R375D substitutions in LeuT (*Figure 4B*) precluded salt bridge formation between R375 and E6, and between R375 and the backbone of I184 as was also observed with Rosetta modeling in *Figure 4A*. In addition, K189 moved towards E6 by 3.4 Å, reducing the distance between residues K189 and E6 from 8.0 Å in LeuT WT to 4.6 Å in LeuT R375A (*Figure 4B*, middle bottom), in agreement with Rosetta modeling. The distance between residues R5 and D369, and between residues E6 and I187, was conserved in all three structures (*Figure 4B*, right bottom), as also found with Rosetta modeling (*Figure 4A*). As evident from these data, as well as the REU versus RMSD plots (*Figure 4—figure supplement 1D*), our Rosetta models parallel our crystal structures. In addition, these data indicate that the IC gate is disrupted by substitutions at position R375

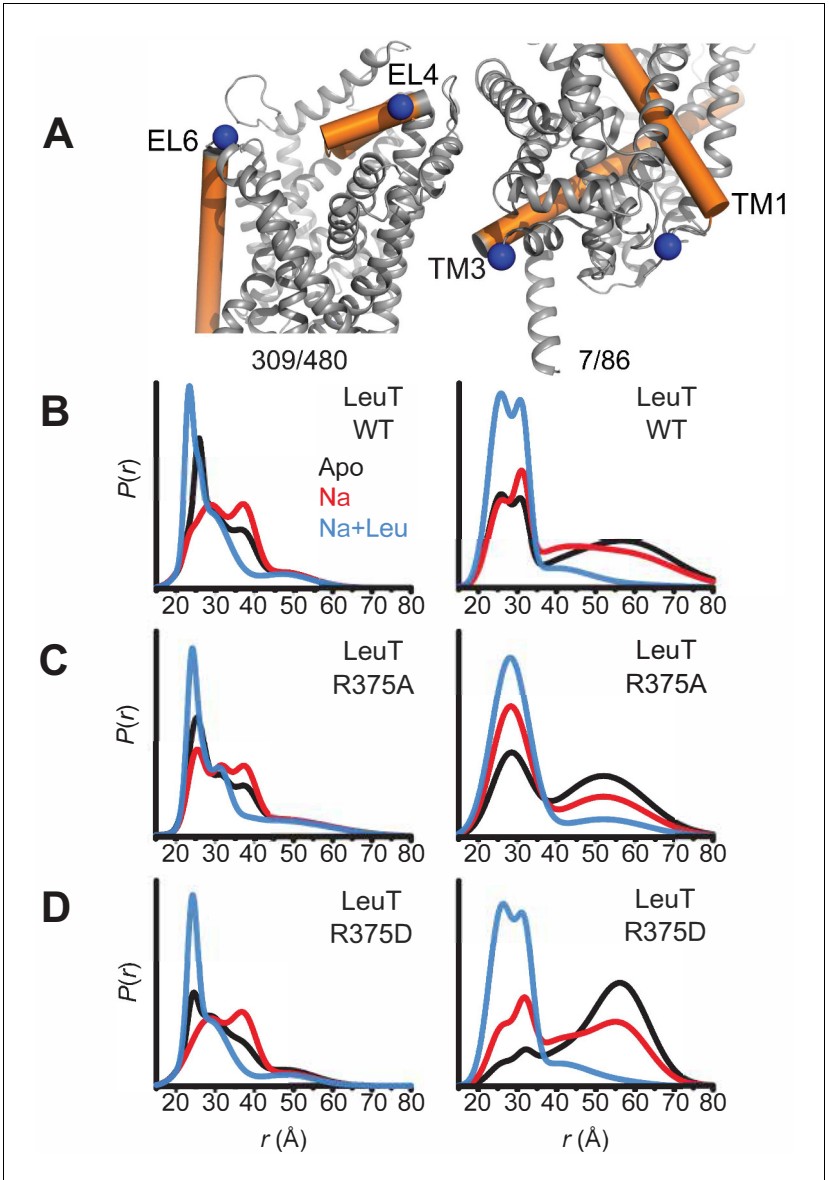

**Figure 5.** Asp substitution at R375 in LeuT favors an inward facing conformation Probability distance distributions (P(r)) of the spin labels 308/409 and 7/86 (**A**) reporting the conformational dynamics of the EC (*left*) and IC (*right*) gate of LeuT, respectively. Distance distributions for each pair were obtained in the Apo (*black*), $Na^+$-bound ($Na^+$; *red*), and $Na^+$- and Leu-bound ($Na^+$/Leu; *blue*) conformations for LeuT WT (**B**), LeuT R375A (**C**), and LeuT R375D (**D**).

as a result of molecular rearrangements more complex than previously hypothesized (*Reith et al., 2018*).

## R375 substitutions disrupt alternating access in LeuT

To monitor impact of R375 substitution on the ligand-dependent conformational dynamics of the EC and IC gates, we used EPR, and more specifically, double electron-electron resonance (DEER), to obtain distance distributions between spin label pairs 309/480 and 7/86 (*Figure 5A*, left and right, respectively). These spin label pairs are used to monitor the isomerization of LeuT between the OF, OO, IF and inward-facing occluded (IO) states, as previously described (*Campbell et al., 2019*; *McHaourab et al., 2011*). It is important to note that the spin labels were attached at introduced cysteines, hence precluding the investigation of LeuT R375C. Instead, we monitored the effects of

R375A and R375D substitutions on LeuT conformational dynamics. We found that these substitutions had relatively minor effects on the EC gate (309/480 pair). In the absence of ion and substrate (Apo), LeuT WT dwells between OO and OF conformation, with OO being predominant (*Figure 5B*, left; *black trace*). Na$^+$ enhances the OF conformation poised to bind substrate (*Figure 5B*, left; *red trace*) (*Claxton et al., 2010*). Leu binding to Na$^+$-bound LeuT restores the conformational preference to the OO form (*Figure 5B*, left; *blue trace*). We found that the introduction of an Ala (*Figure 5C*, left) or Asp (*Figure 5D*, left) at position R375 did not drastically affect the EC gate in the Na$^+$/Leu intermediate (*blue trace*) but, longer-distance components are sampled in the Apo (*black trace*) and Na$^+$ (*red trace*) forms. In R375D, the probability distribution of the dominant short-distance component (OO) decreased, such that more open intermediate distances were sampled in the Apo state (*black trace*) (*Figure 5D*, left).

More substantial changes were observed on the IC gate. Consistent with previous findings, the spin label pair monitoring of the IC gate in LeuT WT showed a bimodal distribution between IF and IO conformations in the Apo state (*black trace*), whereas Na$^+$ alone (*red trace*) begins, and Na$^+$/Leu (*blue trace*) completes, biasing LeuT toward the IO conformer (*Figure 5B*, right). The substitution R375A increased the probability of an IF conformation only in the Apo (*black trace*) and Na$^+$/Leu states (*blue trace*) (*Figure 5C*, right). Similarly, R375D suppressed the short-distance component (IO conformation) in favor of an IF conformation in the Apo state (*black trace*) (*Figure 5D*, right). The addition of Na$^+$ was able to partially rescue the probability distribution of the IO conformer, where Na$^+$/Leu resets the IC gate to the IO conformation (*Figure 5D*, right).

Together, DEER distance distributions demonstrate that the substitution of R375D leads to increased probability of open conformations on both sides of the transporter. This may suggest the population of a channel-like state consistent with the prediction from MD simulations described below.

## R445 substitutions lead to the intermittent formation of a channel-like intermediate in hDAT

To determine the structural and dynamic changes induced by a R445C substitution in hDAT, we generated homology models of hDAT based on dDAT structures (PDB ID: 4M48). As illustrated in *Figure 6A*, salt-bridges at the IC surface (e.g. R445-E428 and R60-D436), a cation-π interaction between R60 and Y335, and a hydrogen bond between E428 and Y335, form an IC network of interactions that stabilizes the occlusion of the IC vestibule in hDAT WT (*Figure 6A*; *Cheng and Bahar, 2015*; *Kniazeff et al., 2008*; *Shan et al., 2011*). In silico studies have suggested that disruption or reconfiguration of these IC salt bridges facilitate the opening of the IC vestibule for release of substrate or ions (*Cheng and Bahar, 2015*; *Khelashvili et al., 2015b*). This feature has also been noted in the human serotonin transporter (hSERT) in recent cryo-EM structures (*Cheng and Bahar, 2019*; *Coleman et al., 2019*).

The structural model generated for hDAT R445C showed that this substitution disrupts this IC interaction network to support an intermittent channel-like intermediate (*Figure 6B*) which is characterized by continuous water occupancy in the transporter lumen. Superposition of hDAT WT and R445C structures (*Figure 6C*) showed an overall opening of the transmembrane (TM) helices on the IC face (TM9, *blue arrow*) in hDAT R445C. MD simulations also showed that Na$^+$ migrates from either the IC or EC side (*Figure 6D*), where Na$^+$ binding occurs prior to the complete dissociation of R60-D436 salt bridge at 150 ns that is paralleled by the formation of the new E428-R60 salt bridge (*Figure 6E*). We also note that the IC-exposed TM1a-TM6b pair retained their 'closed' state (*Figure 6F*), in contrast to the usual opening of TM1a in the IF state observed in WT. Finally, Na$^+$ entry was facilitated by the opening of TM9 and consequent increase in the interhelical distance between TM9 and TM6b (*Figure 6F*).

Similar channel-like intermediates were observed in hDAT R445A (data not shown) and R445D (*Figure 6—figure supplement 1*). In R445D, three Na$^+$ ions (*cyan, violent and orange spheres*) stabilize along the solvated transporter lumen: one entering from the IC region, one entering from the EC region, and one intermittently diffusing from the IC region (*Figure 6—figure supplement 1B*). In contrast, only two Na$^+$ binding sites are present in R445C (*Figure 6B*). It is likely that the dissociation of the R445-E428 salt bridge promoted by R445D substitution allows E428 to bind an additional Na$^+$. In contrast, in hDAT R445C, E428 finds an alternative partner, R60. These findings point to a

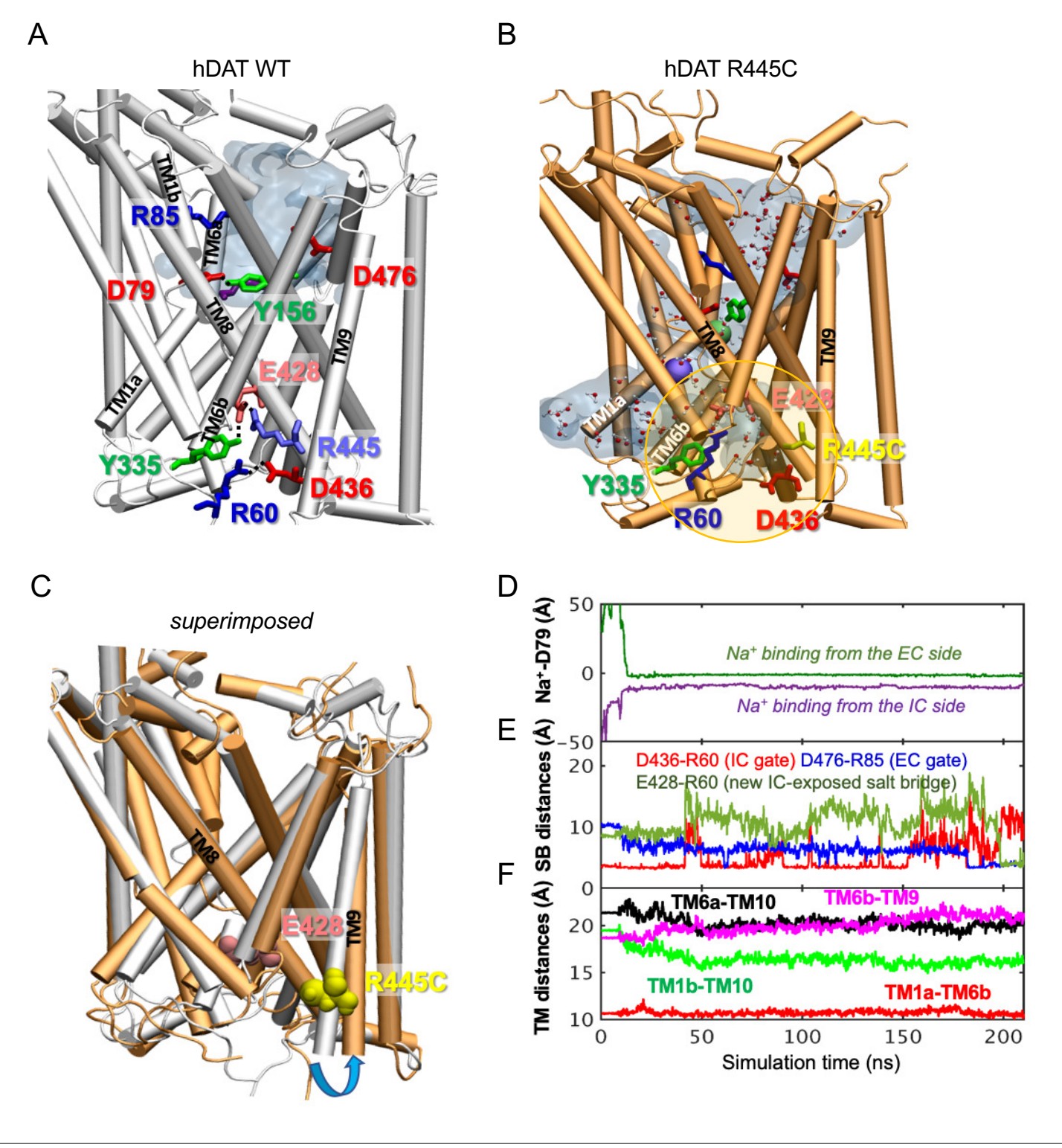

**Figure 6.** hDAT R445C favors the opening of the IC vestibule, leading to intermittent formation of a channel-like intermediate. (**A**) Structure of hDAT WT in the OF state (*white*) illustrates a network of interactions at the IC surface stabilizing the closure of IC vestibule and solvated EC vestibule (*gray shaded region*). (**B**) Substitution of R445 with Cys (*orange*) breaks salt-bridge R445-E428, which weakens IC network interactions and promotes the intermittent formation of a channel-like intermediate. This conformation favors the entry of both water and ions from the IC space. Hydrated regions inside the transporter are indicated in *gray* shaded areas with explicit water molecules displayed in spheres and lines (CPK format). Green and purple spheres are Na+ migrating from the EC and IC side, respectively. (**C**) Structural alignment of hDAT R445C (*orange*) with hDAT WT (*white*). In hDAT

*Figure 6 continued on next page*

*Figure 6 continued*

R445C, the association between TM8 and TM9 (near the IC exposed region) is weakened. TM9 undergoes an outward titling (*blue curved arrow*) to allow for the 'opening' of IC vestibule along TM8, facilitated by the absence of C445-E428 salt bridge (R445-E428 in hDAT WT holds TM8-TM9 in place). (**D–F**) Results from MD simulations of hDAT R445C. Time evolution of distances between $Na^+$ and D79 (**D**); between salt-bridge forming residues at EC and IC regions (**E**) are displayed. On the EC side, D476-R85 distance decreases (EC gate closure). On the IC side, D436-R60 distance increases (IC gate opening). D345-K66 remains closed. After dissociating from D436 (t = 150 ns), R60 interacts with E428 (t = 200 ns). (**F**) Interhelical distances for EC-exposed TM1b-TM10 and TM6a-TM10 shows that the EC region remains exposed to solvent with reduced opening, and IC-exposed TM1a-TM6b is closed, but there is a new opening indicated by the increase in TM6b-TM9 distance. Conformation shown in B is the last snapshot taken from the simulation trajectory in **D–F**.

The online version of this article includes the following figure supplement(s) for figure 6:

**Figure supplement 1.** Related to *Figure 6*.

unique feature of the R445 residue, as substitutions associated with DTDS stabilize a channel-like conformation only observed occasionally in previous simulations (*Cheng et al., 2018*).

We also observe that the dissolution of the R445-E428 salt-bridge weakened the IC interaction network as a whole. In particular, R445C weakened the association of TM8-TM9 near the IC entrance, whereby TM9 underwent an outward tilting exposing an egress pathway along TM8 for $Na^+$ (or a different cation) (*Figure 6C*). The outward titling of TM9 has been observed previously in the DA-loaded transition from OF to IO states (*Cheng and Bahar, 2015*). Furthermore, R455C substitution increases the likelihood that the R60-D436 salt bridge breaks, while promoting the formation of a new salt bridge R60-E428 (*Figure 6E*) at the expense of breaking R60-D436 salt-bridge in both runs. This new salt bridge may lock the IC gate in a new configuration.

In a heterologous expression system, hDAT R445C displays reduced expression that is partially rescued by chloroquine hDAT R445C isolated brains display a reduction in DA content (*Figure 3A*). Unfortunately, we could not consistently measure hDAT expression in this system. Thus, we investigated the expression of hDAT R445C in a heterologous expression system (HEK 293 cells). R445C substitution reduced the surface expression to $0.06 \pm 0.01$ of hDAT WT ($1.0 \pm 0.04$; p<0.0001) and the total mature DAT expression to $0.20 \pm 0.04$ of hDAT WT (marked by #; $1.0 \pm 0.05$; p<0.0001) (*Figure 7A*). Given this reduction in transporter expression, in addition to structural rearrangements, we suspected that DA uptake would also be impaired. Indeed, [$^3$H]DA uptake kinetics showed that R445C expressing cells have significantly reduced transport capacity with respect to WT cells, as reflected in the $V_{max}$ ($F_{(1, 15)}$=160.3; p<0.0001) (*Figure 7B*). However, the apparent affinity for DA ($K_m$) significantly increased in hDAT R445C relative to WT (p<0.0001) cells, suggesting that conformational changes required for translocation of DA across the membrane are also affected (*Figure 7B*). To determine if R445C affected the reverse transport function of the DAT (DA efflux), we delivered DA (2 mM for 10 min) to the inside of the cell through a patch-pipette in whole-cell configuration and used amperometry to measure DA efflux in response to AMPH (10 μM) (*Belovich et al., 2019*). Thus, we were able to load the cells with equal concentrations of DA despite differences in DA uptake. Consistent with our ex vivo brain amperometric recordings, we found that R445C supported DA efflux, albeit significantly reduced compared with WT (hDAT WT = $0.74 \pm 0.09$ pA; hDAT R445C = $0.28 \pm 0.06$ pA; p=0.001) (*Figure 7C*).

We also found that both neutralizing and anionic substitutions at R445 (hDAT R445A and hDAT R445D) significantly compromised surface DAT (p<0.0001) and mature DAT expression (p<0.0001) relative to hDAT WT (*Figure 7—figure supplements 1A* and *2A*). In agreement with this reduction in hDAT surface expression and observed structural impairment, [$^3$H]DA uptake was also significantly reduced in hDAT R445A ($F_{(5,92)}$ = 22.7, p<0.0001; *Figure 7—figure supplement 1B*) and hDAT R445D expressing HEK 293 cells ($F_{(5,94)}$ = 42.1; *Figure 7—figure supplement 2B*). Consistent with data from the R445C mutant, we find that the $K_m$ of hDAT R445A and hDAT R445D was also significantly increased. Combining patch-clamp with amperometry (as above), we found that AMPH-induced DA efflux was significantly compromised in hDAT R445D (p=0.002; *Figure 7—figure supplement 2C*) compared with hDAT WT cells. Interestingly, we observed that AMPH caused a reduction in the amperometric current in hDAT R445A compared with hDAT WT cells (p=0.001; *Figure 7—figure supplement 1C*), consistent with AMPH blocking constitutive DA efflux, as

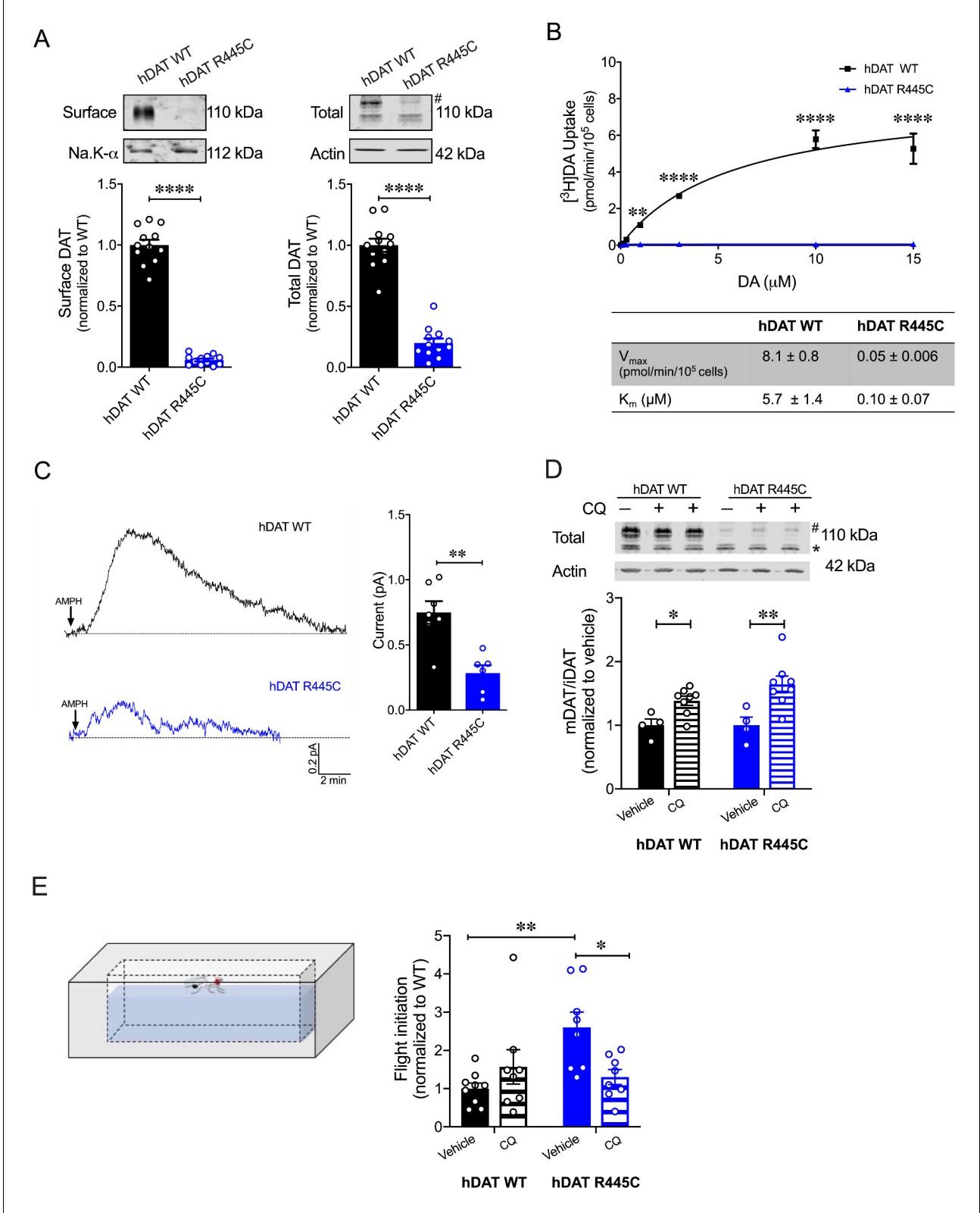

**Figure 7.** CQ enhances R445C expression ratios in transfected HEK 293 cells as well as *Drosophila* flight coordination. (**A**) Representative immunoblots of surface hDAT (top *left*), total hDAT (top *right*) and corresponding Na-K ATPase (*bottom left*) and actin (*bottom right)* loading controls. hDAT expression was normalized to hDAT WT. hDAT R445C displayed significantly reduced surface (p<0.0001; n = 4, in triplicate) and total glycosylated (#) expression relative to hDAT WT (p<0.0001; n = 4, in triplicate). (**B**) Average $^3$[H]DA saturation curves of DA uptake measured in hDAT WT (*black*) or
*Figure 7 continued on next page*

Figure 7 continued

hDAT R445C (*blue*) cells (n = 3, in triplicate). Curves were fit to Michaelis-Menten kinetics to derive $K_m$ and $V_{max}$. DA uptake for hDAT R445C was significantly reduced compared with hDAT WT at every DA concentration measured ($F_{(6,64)}$ = 52.4, p<0.0001), as were the kinetic constants, $K_m$ and $V_{max}$ (p<0.0001). (C) Representative traces of amperometric currents (DA efflux) recorded in response to AMPH application (*left*; 10 μM, indicated by arrow) from hDAT WT (*black*) and hDAT R445C (*blue*) cells loaded with DA (2 mM, 10 min) with whole-cell pipette. Quantitation of peak current amplitudes illustrated a significant reduction in DA efflux recorded from hDAT R445C compare to hDAT WT (*right*; p=0.008; n = 6–7). (D) Representative immunoblots of total hDAT (*top*) and actin loading controls (*bottom),* where glycosylated (#) and non-glycosylated (*) hDAT is highlighted. Ratio of mature (glycosylated) to immature (non-glycosylated) DAT (mDAT/iDAT) expression following CQ treatment was normalized to vehicle condition for hDAT WT and hDAT R445C cells (n = 4, in duplicate). Incubating hDAT R445C cells with CQ (1 mM, 4 hr) significantly increased the ratio of mDAT/iDAT ($F_{(1,20)}$ = 18.0), p=0.003. CQ also significantly increased mDAT/iDAT expression in hDAT WT cells (p=0.04). (E) Diagramed is the flight initiation assay used to determine take-off times for hDAT WT and hDAT R445C *Drosophila* (*left*). hDAT R445C and hDAT WT *Drosophila* were fed CQ (3 mM, 72 hr) or vehicle supplemented diet. Quantitation showed a significant reduction in the time to initiate flight in hDAT R445C flies ($F_{(1,29)}$ = 8.7, p=0.04) in response to CQ compared with vehicle conditions (*right*). Time for flight initiation was comparable in hDAT WT flies exposed to CQ and vehicle supplemented diet (p>0.05). Data represent mean ± SEM. Student's t-test (A) and (C); Two-way ANOVA with Bonferroni's multiple comparison test: (B), and (D–E).

The online version of this article includes the following figure supplement(s) for figure 7:

**Figure supplement 1.** R445A substitution impairs hDAT expression and function in HEK 293 cells.
**Figure supplement 2.** R445D substitution impairs hDAT expression and function in HEK 293 cells.

previously noted in other DAT mutations (*Bowton et al., 2010*; *Mazei-Robison et al., 2008*). Together, these data confirm that, in HEK 293 cells, substitutions at R445 significantly compromised DAT cell surface expression and function.

The severity and onset of clinical phenotypes are associated with residual DAT function in DTDS (*Kurian et al., 2011*; *Ng et al., 2014*). DAT function is related to its expression in hDAT R445C and other DTDS-associated variants; thus, we assessed the possibility of improving motor coordination deficits in hDAT R445C flies by enhancing/correcting DAT expression. DAT expression and degradation are regulated by endocytic, recycling, and lysosomal pathways (*Daniels and Amara, 1999*; *Loder and Melikian, 2003*; *Miranda et al., 2007*; *Wu et al., 2015*). Previous studies have shown that chloroquine (CQ), a lysosomotropic weak base that inhibits lysosomal activity, limits DAT lysosomal degradation (*Cartier et al., 2019*; *Daniels and Amara, 1999*). In DTDS-associated variants, the ratio of mature (glycosylated; mDAT) to immature (unglycosylated; iDAT) DAT is shifted where the immature form predominates (*Figure 7A*; *Kurian et al., 2011*), suggesting DAT degradation. Here, we investigated whether inhibition of DAT lysosomal degradation using CQ could improve this ratio. We found that CQ treatment (1 mM, 4 hr) significantly increased the ratio of mature DAT (marked by #) to immature DAT (marked by *) in hDAT WT (vehicle: 1.0 ± 0.07; CQ: 1.4 ± 0.06; p=0.04) as well as in hDAT R445C-expressing cells (vehicle: 1.0 ± 0.1; CQ: 1.6 ± 0.1; p=0.003) ($F_{(1,20)}$ = 18.0) (*Figure 7D*). As specified above in DTDS, the severity of clinical phenotypes is correlated with DAT function/expression. Thus, we sought to determine whether the improvement in DAT expression promoted by CQ, in a heterologous expression system, translated to in vivo improvements in motor phenotypes. We supplemented fly food with either CQ (3 mM, 72 hr) or vehicle for both hDAT WT and hDAT R445C flies and measured the timing of flight initiation. We found that CQ treatment significantly improved the time for flight initiation in hDAT R445C flies relative to vehicle ($F_{(1,29)}$ = 8.7, p=0.04) (*Figure 7E*). CQ did not have significant effects on motor coordination in hDAT WT flies (p>0.05). These data suggest that CQ, by enhancing DAT expression, can improve flight initiation in hDAT R445C flies, and that when a threshold level of DAT expression is achieved, further increases in DAT expression do not enhance flight initiation time.

## Discussion

PD is a multi-system, heterogenous neurodegenerative disorder characterized clinically by core motor symptoms including resting tremors, bradykinesia, rigidity, and postural instability. As the disease progresses, additional motor symptoms develop, such as impairments in gait and balance, eye movement control, speech and swallowing, and bladder control. Mood disorders (e.g. anxiety and depression), sleep disorders (e.g. insomnia, disrupted circadian rhythm), hyposmia (impaired

olfaction), gastrointestinal symptoms and other non-motor features usually precede full PD diagnosis (*Faivre et al., 2019*). Additionally, cognitive impairment, including dementia, typically manifests after diagnosis and progresses steadily over time (*McGregor and Nelson, 2019*). Some motor and behavioral symptoms can be alleviated by DA replacement therapies, such as levodopa (L-DOPA, a DA precursor), DA metabolism inhibitors and DA receptor agonists (*Jenner, 2015*). However, as the disease progresses, there is often a 'loss of drug' effect, with symptoms largely refractory to therapeutic interventions (*Jenner, 2015*). In some patients, DA replacement can promote new behavioral phenotypes, most commonly: impulse control disorder (ICD) and DA dysregulation syndrome (DDS). In ICD, patients impulsively or compulsively engage in reward-seeking behaviors, including gambling, eating, or sexual activities (*Weintraub et al., 2010*). In DDS, patients display addictive behaviors with dependence or withdrawal-type symptoms towards their DA medications (*Giovannoni et al., 2000*). Essential to developing new pharmacotherapies is understanding the underlying disease pathology.

Although the cause of PD is not completely understood, a combination of aging, neuronal susceptibility, genetic risks, and environmental factors have been found to contribute its etiology. Studies on highly penetrant mutations identified in familial parkinsonism, as well as candidate gene and genome-wide association findings in idiopathic PD, have contributed to our understanding of the molecular mechanisms underlying disease pathology (*Trinh and Farrer, 2013*). DTDS is a distinct type of infantile parkinsonism-dystonia associated with DAT dysfunction that shares various clinical phenotypes with PD, including motor deficits and altered DA homeostasis (*Kurian et al., 2011*; *Kurian et al., 2009*; *Ng et al., 2014*). Investigations on DTDS-associated DAT variants are essential to understanding the impact of DAT dysfunction on DA neurocircuits and signaling pathways. Further, these studies may shed light on the molecular mechanisms that underlie the clinical phenotypes shared by DTDS and PD (*Mou et al., 2020*).

In this study, we define how a specific DAT variant identified in DTDS (R445C) confers DAT dysfunction as well as impairments in DA neurotransmission and associated behaviors. R445C alters the structure and gating dynamics of an IC interaction network, as well as DAT expression, in HEK 293 cells. In the NSS superfamily, which includes DAT and LeuT, thermodynamic coupling of substrate and $Na^+$-co-transport occurs via an alternating access mechanism that comprises the opening and closing of the IC and EC gates (*Beuming et al., 2006*; *Yamashita et al., 2005*). R445 is aligned to R375 in LeuT, which forms the R375-E6 salt-bridge as part of the IC gate (*Cheng and Bahar, 2014*). Our crystallographic data, supported by modeling and $\Delta\Delta G$ calculations in LeuT, revealed that substitutions at R375 in LeuT disrupt key IC interactions, including the R375-E6 salt-bridge, promoting an IF conformation.

These findings are consistent with previous in silico studies which suggest that the transition to an IF *conformation* is defined by the dissolution of IC salt-bridges D369-R5 and R375-E6 in LeuT (*Cheng and Bahar, 2014*). Furthermore, from our EPR studies, we surmise that R375 substitutions disrupt the IC network and bias LeuT to an IF conformation, subsequently altering transport, which requires LeuT to isomerize toward an OF conformation. We have previously shown that other mutations associated with neuropsychiatric disorders (i.e. ΔV269) that disrupt this IC network also bias LeuT to an IF conformer, impairing transporter function (*Campbell et al., 2019*). It is important to note that in the Apo conformation R375A does not alter LeuT IC gate to the extent of R375D. This suggests that some variants at this site are more tolerated, likely due to nearby residues conferring redundant interactions to this IC interaction network. Indeed, it has been previously noted that the microenvironment surrounding the IC gate is enriched with putative interaction partners that reinforce this IC network (*Kniazeff et al., 2008*). Finally, in the EC gate, R375A promotes longer-distance components sampled in the Apo, $Na^+$, and $Na^+$/Leu states.

Using homology modeling and MD simulations, we were able to uncover the structural and dynamic changes induced by the R445C mutation in hDAT. In hDAT, R445 forms a salt bridge with E428, an association that is highly conserved among several eukaryotic NSS members and is proposed to be part of the IC gate (*Reith et al., 2018*). Although this association is distinct from the R375-E6 salt-bridge in LeuT, it is thought to serve similar functions as part of the IC network. We found that R445C promotes the dissociation of salt-bridge R445-E428, as previously predicted by Reith and collaborators by using molecular graphics (*Reith et al., 2018*). However, our MD simulations demonstrate that, unexpectedly, the R445C mutation also disrupts the R60-D436 salt bridge and induces intermittent formation of a new salt bridge, E428-R60. These rearrangements of the IC

network give rise to a channel-like intermediate filled with water molecules. This channel-like intermediate was also observed in hDAT R445D, with an additional $Na^+$ binding the transporter from the IC environment. Previous studies have shown that DAT undergoes uncoupled DAT-mediated ionic fluxes (*Ingram et al., 2002*), as well as reverse transport of DA (efflux), via channel-like pathways (*Kahlig et al., 2005*). We have previously uncovered that the hDAT coding variant A559V, identified in patients with ADHD, supports a channel-like mode in DAT which is associated with persistent DAT-mediated reverse transport of DA (DA leak) uncovered by AMPH blockade (*Bowton et al., 2014*; *Mazei-Robison et al., 2008*). This DA leak was also identified in hDAT T356M, a de novo missense mutation in ASD (*NIH ARRA Autism Sequencing Consortium et al., 2013*). We conclude that the channel-like intermediate observed in our simulations of R445 substitutions may be associated with a channel-like mode supporting ion fluxes. Interestingly, we found that a neutral substitution at R445 (hDAT R445A) results in constitutive, anomalous DA leak blocked by AMPH. These data highlight the possibility that anomalous DA efflux may increase risk for various psychiatric disorders (*Bowton et al., 2010*; *NIH ARRA Autism Sequencing Consortium et al., 2013*; *Hansen et al., 2014*; *Mazei-Robison et al., 2008*). It is important to note that constitutive DA efflux is not observed in cells expressing hDAT R445C nor hDAT R445D. This underscores the complexity of the IC network and the possibility that distinct amino acid substitutions at R445 differentially affect the IC dynamics, promoting different hDAT functions.

Our in vitro analysis, combined with our in silico data, revealed that impaired DAT R445C transport capacity stems both from a reduction in transporter expression as well as impaired hDAT function that reflects compromised DA uptake, but partially supported DA efflux. This decrease in DAT function could affect dopamine neuron excitability (*Ingram et al., 2002*). Also, our findings are consistent with previous studies highlighting impaired transporter expression and uptake in hDAT R445C cells (*Asjad et al., 2017*; *Beerepoot et al., 2016*; *Ng et al., 2014*). In addition, these data support the idea that the IC gate differentially regulates inward versus outward transport of DA (*Campbell et al., 2019*), as R445C supports DA efflux (albeit reduced). In addition, we find that specific substitutions at this IC interaction network are distinctly tolerated in LeuT versus hDAT, as has been previously noted with other substitutions at this site (*Stolzenberg et al., 2015*). In hDAT, neither neutral (R445A) nor acidic (R445D) substitutions support normal hDAT function. These findings contrast EPR measurements, which suggest that neutral (R375A) but not acidic (R375D) substitutions are tolerated in LeuT. These findings highlight key differences in the IC and perhaps redundancy existing in LeuT within this network that is absent in the hDAT.

Despite neuroanatomical differences between mammalian and fly brains, increasing evidence on the evolutionary relationships between molecules, neural networks and organization within mammalian and invertebrate brains, as well as studies on animal models of disease, suggest many similarities (*Anderson and Adolphs, 2014*; *Feany and Bender, 2000*; *Hartenstein and Stollewerk, 2015*; *Kaiser, 2015*; *Nagoshi, 2018*; *Xiong and Yu, 2018*; *Yamamoto and Seto, 2014*). We used *Drosophila* as an animal model to explore the physiological and phenotypic consequences of a cysteine substitution at R445 of DAT. Our studies found that hDAT R445C promotes altered motor and coordinated behaviors in *Drosophila*. Specifically, hDAT R445C *Drosophila* displayed impaired locomotion that was driven by compromised movement vigor (fast movement). This behavioral phenotype is parallel to bradykinesia observed in patients with DTDS and PD (*Chai and Lim, 2013*; *Kurian et al., 2011*; *Kurian et al., 2009*; *Ng et al., 2014*) as well as in various mammalian and *Drosophila* models of PD (*Feany and Bender, 2000*; *Nagoshi, 2018*; *Taylor et al., 2010*).

In patients with PD, loss of DA neurons elicits impaired movement and motor symptoms, as well as compromised fine and gross motor coordination. In flies, flight initiation requires exquisite sensory-motor integration. A fly first raises its wings to a ready position, and then subsequently, extends its mesothoracic legs and depresses its wings simultaneously to coordinate a jump with the initial downstroke (*Card and Dickinson, 2008*). Here, using a high-speed camera, we studied flight initiation in a *Drosophila* model of DTDS to understand the effects of hDAT R445C on sensory-motor integration. We found spontaneous flight initiation to be significantly delayed in *Drosophila* expressing hDAT R445C. Similarly, recent studies have observed wing coordination defects in flies with reduced neurotransmitter release from DAergic neurons (*Sadaf et al., 2015*). These data suggest that there may be a reduction in DA tone in hDAT R445C flies that contributes to flight deficits. In addition, we found a disparate repetitive motor behavior that requires fine-motor coordination, grooming, to be unaffected in hDAT R445C *Drosophila*. These findings point to the phenotypic

heterogeneity commonly observed in DTDS and PD (*Faivre et al., 2019*; *Kurian et al., 2011*; *Kurian et al., 2009*; *Lill, 2016*; *Ng et al., 2014*; *Trinh and Farrer, 2013*) and suggest that specific coordinated movements are impaired or alternatively, may present in a progressive nature in this model of DTDS. It is important to note that this study focused on the motor symptoms exhibited in DTDS and PD. However, whether and if R445C promotes non-motor deficits, including cognitive impairment and hyposmia, has not been explored.

Our studies in fly brains also demonstrate that the hDAT R445C mutation drives decreased transporter function (i.e. DA efflux), impaired DA synthesis and reduced TH-labeled DA neurons. Our findings of diminished transporter function align with our previous in vitro findings showing reduced hDAT R445C expression (*Ng et al., 2014*). However, in our preparation, we detected a robust decrease in hDAT R445C. This is in contrast to what has been reported previously using a different cell line, where the hDAT R445C expression was only slightly decreased (*Ng et al., 2014*). Furthermore, we observe an increase affinity for DA in the hDAT R445C mutant. Instead, Ng and collaborators reported that this mutation decreased affinity for DA, as determined by measuring the Ki of dopamine inhibition of a cocaine-analogue. In PD, core motor deficits are ascribed to the loss of DA neurons in the substantia nigra and their projections to the striatum (*Trinh and Farrer, 2013*). To date, neurodegeneration in DTDS has not been studied in depth; however, given that affected individuals develop parkinsonism-dystonia, including resting and acting tremor, difficulty initiating movements, bradykinesia and rigidity, it is likely that DA circuits are affected. To this end, we observed a reduction in TH-labeled DA neurons in hDAT R445C flies, consistent with various *Drosophila* models of PD that show selective neurodegeneration of protocerebral posterior lateral 1 (PPL1) DA neurons (*Barone et al., 2011*; *Cackovic et al., 2018*; *Trinh et al., 2008*; *Whitworth et al., 2005*). Although these findings suggest neurodegeneration in PPL1 neurons, given the reduction in measured DA levels, it is also possible that there is an overall reduction in TH, which limits the labeling of this neuronal population. However, our behavioral data point to a decrease in DA function, as observed in PD.

Together, these findings support a possible mechanism where reduced DAT-mediated DA reuptake results in excessive EC dopamine and depleted presynaptic stores. Synaptic hyperdopaminergia leads to overstimulation of presynaptic $D_2$ autoreceptors which suppress DA release and down-regulate tyrosine hydroxylase (TH), the rate limiting enzyme in DA synthesis, thereby, decreasing DA synthesis (*Ford, 2014*). However, neither level of extracellular DA or TH function/expression were determined in this study. This possible mechanism aligns with previous findings in DAT knockout animals and those with compromised DAT function (*DiCarlo et al., 2019*; *Jones et al., 1999*; *Salvatore et al., 2016*).

DTDS presents in a phenotypic continuum, where clinical phenotypes appear to be associated with varied, residual DAT function. Thus, higher residual DAT activity is suggested to reduce symptom severity and/or postpone the age of disease onset (*Kurian et al., 2011*; *Ng et al., 2014*). Previous studies have used pharmacological chaperones that stabilize the DAT in an IF conformation to rescue transporter expression (*Asjad et al., 2017*; *Beerepoot et al., 2016*; *Ng et al., 2014*). hDAT R445C function was not consistently rescued with these agents (*Asjad et al., 2017*; *Beerepoot et al., 2016*), in alignment with our EPR and in silico data, which showed that hDAT R445C can isomerize and is even biased toward the IF conformer. In light of previous studies which showed that CQ inhibits DAT lysosomal degradation (*Cartier et al., 2019*; *Daniels and Amara, 1999*), we tested CQ for its ability to improve motor deficits in hDAT R445C flies. We found that CQ was able to increase motor coordination in hDAT R445C flies, reducing the time to initiate flight significantly. This improvement in flight coordination was associated with improved DAT expression observed in a heterologous expression system. However, it is important to consider that DA neurons in the PPL1 cluster promote wakefulness in *Drosophila* (*Liu et al., 2012*) and their dysfunction could also contribute to the phenotypes observed in the hDAT R445C flies. It is important to note that in some studies, lysosomal dysfunction has been associated with PD (*Chai and Lim, 2013*; *Trinh and Farrer, 2013*). In these instances, lysosomotropic agents should not be considered, as they may exacerbate disease progression. In addition, although CQ and other quinines have been used for more than 400 years to treat malaria and more recently, re-purposed to treat cancer, these agents are not without substantial adverse side effects (*Achan et al., 2011*; *Weyerhäuser et al., 2018*). Thus, the use of lysosomotropic agents, such as CQ, should be considered as therapeutic agents to ameliorate motor deficits only in specific cases of DTDS.

Our study reveals how a specific DAT variant identified in DTDS contributes to DAT dysfunction and subsequently, how DAT dysfunction supports altered DA neurotransmission as well as behaviors in *Drosophila*. Moreover, this experimental paradigm supports *Drosophila* as a model system in the study of DTDS, and PD, more broadly. Our investigation on hDAT R445C provides a blueprint to gain valuable insights into the mechanisms regulating transporter function, gating and expression, and how dysfunction of these processes translates to abnormal DA physiology and behaviors.

# Materials and methods

## Key resources table

| Reagent type (species) or resource | Designation | Source or reference | Identifiers | Additional information |
|---|---|---|---|---|
| Gene (*Homo sapiens*) | *SLC6A3* | Uniprot | Uniprot Q01959 | Encodes hDAT protein |
| Gene (*Aquifex aeolicus*) | *LeuT* | Uniprot | Uniprot O67854 | Encodes LeuT protein |
| Strain, strain background (*Escherichia coli*) | C43 DE3 F– ompT gal dcm hsdSB (rB- mB-)(DE3) | Lucigen | | |
| Genetic reagent (*D. melanogaster*) | W[1118] | Bloomington Stock Center | BI 6326 | |
| Genetic reagent (*D. melanogaster*) | TH-GAL4 | Bloomington Stock Center | BI 8848 | |
| Genetic reagent (*D. melanogaster*) | DAT[MB07315] | Bloomington Stock Center | BI 25547 | |
| Genetic reagent (*D. melanogaster*) | UAS-mCherry | Kyoto Stock Center | 109594 | |
| Genetic reagent (*D. melanogaster*) | M[vas-int.Dm]ZH-2A; M[3xP3-RFP.attP']ZH-22A | Rainbow Transgenic Flies | 24481 | |
| Genetic reagent (*D. melanogaster*) | *DAT[fmn]* | Gift from Dr. K. Kume | | |
| Cell line (*Homo sapiens*) | HEK | | | |
| Antibody | Anti-DAT (Rat monoclonal) | Millipore | MAB369 | WB (1:1000) |
| Antibody | Anti- β-actin (Mouse monoclonal) | Sigma-Aldrich | A5441 | WB (1:5000) |
| Antibody | Anti- Na-K ATPase | Developmental Studies Hybridoma Bank | a5 | WB (1:100) |
| Antibody | Anti- TH | Millipore | AB152 | IHC (1:200) |
| Antibody | Anti- Bruchpilot | Developmental Studies Hybridoma Bank | nc82 | IHC (1:50) |
| Recombinant DNA reagent | pEGFP (plasmid) | Clonetech | | |
| Recombinant DNA reagent | pET16b (plasmid) | Novagen | | |
| Peptide, recombinant protein | LeuT | This paper | | Purified from C43 (DE3) *E. coli* |
| Peptide, recombinant protein | LeuT R375A | This paper | | Purified from C43 (DE3) *E. coli* |
| Peptide, recombinant protein | LeuT R375C | This paper | | Purified from C43 (DE3) *E. coli* |
| Peptide, recombinant protein | LeuT R375D | This paper | | Purified from C43 (DE3) *E. coli* |

*Continued on next page*

*Continued*

| Reagent type (species) or resource | Designation | Source or reference | Identifiers | Additional information |
|---|---|---|---|---|
| Chemical compound, drug | [³H]dopamine | PerkinElmer Life Sciences | NET673250UC | |
| Chemical compound, drug | Sulfo-NHS-SS-biotin | Fisher | PG82078 | |
| Other | Fugene-6 | Roche Molecular Biochemicals | | |

## Cell culture

HEK cells were authenticated by STR profiling. Cultures were verified to be free of mycoplasma by Mycoplasma Detection Kit (Invivogen rep-pt1) per manufacture's protocol. peGFP expression vector was engineered to contain synhDAT WT (hDAT WT), hDAT R445C, hDAT R445D and hDAT R445A. All vectors were sequenced via Sanger sequencing to confirm mutations. Vector DNA was transiently transfected into human embryonic kidney (HEK) cells using Fugene-6 (Roche Molecular Biochemicals) transfection reagent. eGFP (enhanced green fluorescence protein) was used for cell selection and quantitation of transfection efficiency. Cells were maintained in a 5% $CO_2$ incubator at 37°C in Dulbecco's Modified Eagle Medium (DMEM) supplemented with 10% fetal bovine serum (FBS), 1 mM L-glutamine, 100 U/mL penicillin, and 100 µg/mL streptomycin. All assays were conducted ~48 hr post transfection.

## [³H] DA uptake assays

For DA uptake in a heterologous expression system: Cells were washed in KRH buffer composed of (in mM): 130 NaCl, 25 HEPES, 4.8 KCl, 1.2 $KH_2PO_4$, 1.1 $MgSO_4$, 2.2 $CaCl_2$, 10 d-glucose, 1.0 ascorbic acid, 0.1 pargyline, and 1.0 tropolone. KRH was titrated to pH 7.3–7.4. Cells were equilibrated in KRH at 37°C for 5 min. Saturation kinetics of DA were measured by incubating cells in a range of 0.1 to 15 µM DA, comprised of a mixture of [³H]DA (PerkinElmer Life Sciences, Waltham, MA) and unlabeled DA. Uptake was terminated after 10 min by washing cells twice in ice-cold KRH buffer. Nonspecific binding was measured in the presence of 10 µM cocaine. $K_m$ and $V_{max}$ values were derived by fitting Michaelis-Menten kinetics to specific binding data. For DA uptake in dissected *Drosophila* brains: 2- to 5-day-old males were collected, anesthetized with $CO_2$, and brains were dissected in Schneider's medium (GIBCO) with 1.5% BSA. The retina was removed, and four brains per condition were pooled in Millipore Millicell inserts in 24-well plates. Brains were washed with Schneider's medium, then washed in a standard fly saline solution (HL3) plus 1.5% BSA and 10 mM $MgSO_4$. For 15 min at room temperature, brains were exposed to 200 nM [³H]DA in HL3 plus 1.5% BSA and 115 µM ascorbic acid. Brains were then washed six times with 1.4 mL HL3 plus 1.5% BSA at 4°C. Brains were placed into scintillation vials in 100 µL 0.1% SDS. Scintillation fluid was added to count [³H]DA. Nonspecific binding was measured in the presence of 20 µM cocaine.

## Amperometry and patch-clamp electrophysiology

Cells were washed twice with 37°C Lub's external solution composed of (in mM): 130 NaCl, 1.5 $CaCl_2$, 0.5 $MgSO_4$, 1.3 $KH_2PO_4$, 10 HEPES and 34 d-glucose (pH 7.3–7.4; 300–310 mOsms/L). To intracellularly load DA, a programmable puller (Model: P-2000; Sutter Instruments; Novato, CA) was used to fabricate quartz patch-pipettes with a resistance of 3–8 mΩ. Pipettes were filled with an internal solution containing (in mM): 110 KCl, 10 $NaCl_2$, 2 $MgCl_2$, 0.1 $CaCl_2$, 1.1 EGTA, 10 HEPES, 30 d-glucose, and 2.0 DA (pH 7.3–7.4; 280–290 mOsms/L). Upon gaining whole-cell access, the internal solution was allowed to diffuse for 10 min. To record DA efflux, a carbon fiber electrode was juxtaposed to the plasma membrane of the cell and held at +600 mV. After establishing a baseline, 10 µM AMPH was added to the bath. Amperometric currents were low pass filtered at 1 Hz (Model: 3382; Krohn-Hite Corporation; Brockton, MA), sampled at 100 Hz (Model: Axopatch 200B; Molecular Devices; San Jose, CA), and analyzed off-line using pCLAMP nine software (Molecular Devices). DA efflux was quantified as the peak of the amperometric current.

## Biotinylation assays

Cells were washed on ice with 4°C phosphate-buffered saline (PBS) supplemented with 0.9 mM $CaCl_2$ and 0.49 mM $MgCl_2$. Cells were incubated in 1.0 mg/ml sulfosuccinimidyl-2-(biotinamido) ethyl-1,3-dithiopropionate-biotin (sulfo-NHS-SS-biotin; Pierce, Rockford, IL) in PBS for 20 min at 4°C. Excess biotin was quenched by incubating cells in 100 mM glycine in PBS for 15 min. Cells were solubilized in radioimmunoprecipitation assay buffer (RIPA) composed of 150 mM NaCl, 1.0% NP-40, 0.5% Sodium Deoxycholate, 0.1% SDS, 50 mM Tris, 1 mM EDTA, 1 mM EGTA, 1 mM PMSF and protease inhibitors (1:100), and titrated to pH 7.4. Cellular extracts were centrifuged for 30 min at 16,000 × g at 4°C. The supernatant was added to immunopure immobilized streptavidin beads (Pierce Chemical Company; Rockford, IL) and incubated overnight at 4°C. Beads were extensively washed and eluted in sample buffer. Samples were processed according to a standard western blot protocol (see below).

## Western blotting protocol

Cells were incubated in vehicle or 1 mM chloroquine (CQ) for 4 hr. Cells were solubilized in RIPA, sonicated, and centrifuged. Supernatants were denatured in sample buffer, run on SDS-PAGE gel, and transferred to polyvinylidene fluoride membrane (PVDF) (Millipore, Bedford, MA). Membranes were immunoblotted for DAT (1:1000) (MAB369; Millipore), β-actin (1:5000) (A5441; Sigma-Aldrich; St. Louis, MO), and Na-K ATPase (1:100; Developmental Studies Hybridoma Bank (DSHB), Iowa City, Iowa). The secondary antibodies used were Li-COR goat anti-rat IRDye 800 (1:15,000), goat anti-rabbit IRDye 680 (1:15,000) and goat anti-mouse IRDye 680 (1:15,000). Band densities were quantified using Image Studio Odyssey Infrared Imaging System (LI-COR, Lincoln, Nebraska).

## *Drosophila* rearing and stocks

All *Drosophila melanogaster* strains were grown and maintained on standard cornmeal-molasses media at 25°C under a 12:12 hr light-dark schedule. Fly stocks include $w^{1118}$ (Bloomington Indiana Stock Center (BI) 6326), TH-GAL4 (BI 8848), $DAT^{MB07315}$ (BI 25547), UAS-mCherry (Kyoto Stock Center 109594), and M[vas-int.Dm]ZH-2A; M[3xP3-RFP.attP']ZH-22A (BI 24481) and $DAT^{fmn}$ (dDAT KO). *Drosophila* expressing homozygous dDAT null allele $DAT^{fmn}$ (dDAT KO) (*Kume et al., 2005*), TH-Gal4 (*Friggi-Grelin et al., 2003*), and UAS-mCherry were outcrossed to a control line ($w^{1118}$) for 5–10 generations and selected by PCR or eye color. Transgenes (hDAT WT and hDAT RT445C) were cloned into pBID-UASC (*Wang et al., 2012*) and constructs were injected into embryos from M[vas-int.Dm]ZH-2A, M[3xP3-RFP.attP']ZH-22A (BI 24481) (Rainbow Transgenic Flies Inc; Camarillo, CA). Initial potential transformants were isolated and selected. Flies containing transgenes were outcrossed to dDAT KO flies (in $w^{1118}$ background) for 5–10 generations. Age-paired adult male flies (10 days post eclosion) containing a single copy of hDAT WT or hDAT R445C in DA neurons in a $DAT^{fmn}$ background were used for all subsequent experiments.

## *Drosophila* amperometry assays

 *Drosophila* brains were dissected with surgical forceps in ice-cold Schneider's *Drosophila* Medium supplemented with 1.5% BSA. Whole brains were placed in a mesh holder in Lub's external solution (see previous). A carbon fiber electrode was held at +600 mV and inserted juxtaposed to TH-positive PPL1 DA neuronal region. After establishing a baseline, 20 µM AMPH was added to the bath. Amperometric currents were processed as stated above.

## *Drosophila* locomotion analysis

Spontaneous locomotor activity in an open field was measured using custom 3D printed activity chambers (1.1 × 1.1 cm). Locomotion was detected using NIS Elements AR (Melville, NY). Animals were placed in the activity chambers, where activity was recorded for 5 min following 2 min acclimation period. Data from this test was also used to measure anxiety-like behaviors. Thigmotaxis, the tendency of an animal to remain close to the walls of an open field, was measured as the percent of time flies spent in center square (3.0 × 3.0 mm). Total distance traveled, center time, and velocity distribution were quantified using MATLAB 2018b (MathWorks; Natick, MA). Velocity thresholds for movement initiation were set based on the average velocity ($\chi$) during non-movement phases

($\chi$ +0.5$\sigma$ = 0.50 + 0. 24 mm/s), whereas fast movement was determined from the average velocity during the test period ($\chi$ + $\sigma$ = 2.7 + 2.6 mm/s).

## *Drosophila* grooming analysis

Flies were observed for a period of 5 min (~19 fps). Forelimb and hindlimb grooming incidents were quantified per frame, where total grooming time was calculated as the total number of frames spent grooming.

## *Drosophila* flight assay

Coordinated flight was measured using custom 3D-printed chambers (3.9 × 1.0×1.0 cm) filled with 2600 µL of water. Flight initiation was recorded at 2000 frames per second using a Phantom v1212 Camera (Ametek; Wayne, New Jersey), after a short acclimation period. Delay in flight initiation was quantified as the time from the outset of the first wing motion to the coordinated jump response.

## HPLC

Biogenic amines were quantified by the Neurochemistry Core Facility at Vanderbilt University. Briefly, *Drosophila* brains were dissected quickly in ice-cold PBS and immediately frozen in liquid nitrogen. Brains were homogenized using a tissue dismembrator in 100–750 µl of solvent containing (in mM) 100 TCA, 10 Na, 0.1 EDTA and 10.5% methanol (pH 3.8). Homogenate was spun (10,000 x g, 20 min) and supernatant was removed for biogenic monoamines analysis. Biogenic amine concentrations were determined utilizing an Antec Decade II (oxidation: 0.65) electrochemical detector operated at 33°C. Supernatant was injected using a Water 2707 autosampler onto a Phenomenex Kintex C18 HPLC column (100 × 4.60 mm, 2.6 µm). Biogenic amines were eluted with a mobile phase 89.5% of solvent (see previous) and 10.5% methanol (pH 3.8). Solvent was delivered at 0.6 ml/min using a Waters 515 HPLC pump. Biogenic amines elute in the following order: Noradrenaline, Adrenaline, DOPAC, Dopamine, 5-HIAA, HVA, 5-HT, and 3-MT. HPLC control and data acquisition are managed by Empower software. Isoproterenol (5 ng/mL) was included in the homogenization buffer for use as a standard to quantify the biogenic amines. Protein concentration was determined by BCA Protein Assay Kit (ThermoFisher Scientific).

## Immunohistochemistry

Fly brains were dissected in PBS and fixed in 4% paraformaldehyde for 20 mins at RT. Brains were washed three times with PBST (0.3% Triton X100). Brains were blocked in 1% BSA and 5% normal goat serum. Brains were immunostained for TH (1:200; Millipore, AB152) and nc82 (1:50; DSHB) overnight at 4°C, washed and stained with secondary antibodies Alexa 488–conjugated goat anti-rabbit (1:200; A11034, ThermoFisher Scientific) and Alexa 566–conjugated goat anti-mouse (1:200, A11031, ThermoFisher Scientific) overnight at 4°C. Brains were washed and mounted with ProLong Diamond Anti-Fade mounting solution (ThermoFisher Scientific). Imaging was performed using a Nikon A1R confocal microscope. The resolution of the image stack was 1024 × 1024 with 0.5 µm step size. Neurons were counted manually using FIJI (Bethesda, MD). Data were analyzed blinded to genotype.

## Rosetta homology modeling and stability calculations

The Rosetta Flex $\Delta\Delta$G protocol (*Barlow et al., 2018*; *Kuenze et al., 2019*) and the Rosetta Membrane all-atom energy function (*Alford et al., 2015*) were used to estimate free energy changes and sample conformational changes of the LeuT, hDAT and corresponding variants. The Flex $\Delta\Delta$G protocol models mutation-induced conformational and energetic changes through a series of 'backrub' moves of the protein backbone together with side-chain repacking around the mutation site. 15,000 backrub steps were used in this study to sample backbone and side chain degrees of freedom for neighboring residues within an 9 Å boundary of the mutation site. This is subsequently followed by side chain optimization using the Rosetta 'packer.' Global minimization of the backbone and side chains torsion angles is performed with harmonic C$\alpha$ atom-pair distance restraints. The restraints are used to prevent large structural deviations from the input model. Models are scored with the Rosetta Membrane all-atom energy function (*Alford et al., 2015*). This is carried out in parallel for the WT input model and the mutant of interest. For the LeuT calculations, the LeuT crystal structure

(PDB ID: 2A65) (*Yamashita et al., 2005*) was used and 1000 independent trajectories were carried out for both LeuT WT (control) and each variant. For the hDAT calculations, homology models for hDAT WT were created in the Rosetta molecular modeling suite (revision 57712, Rosetta Commons) as previously described (*Campbell et al., 2019*) using the *Drosophila melanogaster* DAT (PDB ID: 4XP9) (*Wang et al., 2015*) as a structural template. A total of 500 independent trajectories were carried out for each hDAT R445 mutant and hDAT WT (control). This protocol was used for the top three scoring hDAT homology models resulting in 1500 trajectories total per mutant. The Rosetta energy change ($\Delta\Delta G$) was calculated as score difference between the average of the top 5% of LeuT WT and corresponding variants, as well as of hDAT WT and corresponding variants. Rosetta $\Delta\Delta G$ values are in Rosetta Energy Units (REU). Representative structural models for LeuT, hDAT, and all variants were selected for visualization in Pymol by removing outliers and taking the lowest-energy model within the lowest interquartile range of a box plot.

## Protein expression and purification

*Escherichia coli* C41 (DE3) cells were transformed with the pET16b plasmid containing LeuT, LeuT R375A, LeuT R375C, or LeuT R375D tagged with a C-Terminal 8xHis-tag and thrombin cleavage site. Transformed cells were grown in Terrific broth media to an $OD_{600}$ of 0.6. Cells were induced with 0.1 mM isopropyl-β-D-1-thiogalactopyranoside (20 hr, 20°C), harvested by centrifugation and disrupted with a french press in 20 mM HEPES-Tris pH 7.5, 190 mM NaCl, 10 mM KCl, 1 mM EDTA, 5 mM L-Alanine, 100 μM AEBSF, and 0.004 mg/mL DNAse I. Cells membranes were isolated by ultracentrifugation at 200,000 x g (45 min) and solubilised with 40 mM n-dodecyl-β-D-maltopyranoside (DDM, Anatrace). Solubilized membranes were incubated with Ni-NTA resin (Qiagen) (1 hr, 4°C). Protein bound to the Ni-NTA resin was washed with 50 mM imidiazole and then eluted with 300 mM imidiazole. The histidine tag was subsequently removed by digestion with thrombin (10 U/mg protein) and the protein further purified on a size exclusion column in 10 mM Tris-HCl pH 8.0, 45 mM NaCl, 5 mM KCl, 5 mM L-Alanine, and 40 mM n-Octyl-β-D-glucopyranoside (OG, Anatrace). Purified protein was concentrated to 8 mg/mL using 30 kDa cut-off AMICON concentrators (Merck).

## Crystallography and structure determination

Crystals were grown at 18°C using the hanging-drop vapor diffusion method, by mixing protein (~8 mg/ml) and well solution (1:1 vol:vol), 100 mM HEPES-NaOH pH 7–7.5, 200 mM NaCl, 17–22% PEG550 MME. Protein crystals were cryoprotected by soaking in the well solution supplemented with 25–35% PEG550 MME. All diffraction data was collected on the EIGER 16M detector at the Australian Synchrotron (ACRF ANSTO) beamline MX2 at a wavelength of 0.954 Å (*Aragão et al., 2018*). Datasets were indexed, integrated and scaled using XDS (*Kabsch, 2010*). Initial phases were obtained by molecular replacement with Phaser (*McCoy et al., 2007*) using the structure of LeuT with bound L-Leu (PDB ID: 3F3E) as the search model. The protein model was built manually in Coot (*Emsley et al., 2010*) and refined using REFMAC (*Murshudov et al., 2011*) with TLS and non-crystallographic symmetry (NCS) restraints (*Winn et al., 2001*). Phases were further improved by rounds of manual rebuilding followed by restrained refinement in REFMAC. Validation was carried out using MolProbity (*Chen et al., 2010*). Unit cell parameters, data collection, and refinement statistics are presented in *Figure 4—figure supplement 2*. All structural figures were prepared using USCF Chimera (*Pettersen et al., 2004*).

## Electron paramagnetic resonance (EPR) protocol

Cysteine residues were introduced using site directed mutagenesis into LeuT, LeuT R375A, and LeuT R375D constructs. Experiments were conducted as in *Claxton et al., 2010*. The apo conformation refers to $Na^+$ and leucine-free transporter, while the +Na/Leu state was obtained in 200 mM NaCl and fourfold molar excess of Leu relative to LeuT. Double Electron Electron Resonance (DEER) (*Jeschke and Polyhach, 2007*) was performed at 83K on a Bruker 580 pulsed EPR spectrometer operating at Q-band frequency using a standard 4-pulse sequence (*Zou and Mchaourab, 2010*). DEER echo decays were analyzed to obtain distance distributions (*Jeschke et al., 2002*).

## Statistical methods

Experiments were designed using statistical power calculations considering means and standard errors from preliminary data. Statistical analyses were performed using GraphPad Prism 8 (San Diego, CA). Shapiro-Wilk normality tests were performed to determine if data was normally distributed and F tests were performed to compare variances; parametric or non-parametric tests with appropriate corrections were chosen accordingly. All data was acquired unblinded, but analyzed blinded to genotype.

## Molecular dynamics (MD) simulations

The structural model for *apo* hDAT (residues Q58-D600) in the outward-facing open (OF) unbound state, based on dDAT structure (PDB ID: 4M48), was taken from previous study (*Cheng et al., 2018*). Four simulation systems using this initial structure were constructed: wild-type (WT), R445C, R445A, and R445D. In each case, the transporter is embedded into 1-palmitoyl-2-oleoyl-sn-glycero-3-phosphocholine (POPC) membrane lipids using CHARMM-GUI Membrane Builder module (*Wu et al., 2014*). TIP3P waters and $Na^+$ and $Cl^-$ ions corresponding to 0.15 M NaCl solution were added to build a simulation box of ~110 × 110×118 Å. Each simulation system contained ~131,000 atoms, the transporter, ~300 lipid molecules, and 27,000 water molecules. All simulations were performed using NAMD (*Phillips et al., 2005*) (version NAMD_2.12) following previous protocol (*Cheng et al., 2018*). For each mutant, two independent runs of 200 ns are performed to verify the reproducibility of the results. VMD (*Humphrey et al., 1996*) with in-house scripts was used for visualization and trajectory analysis.

# Acknowledgements

The authors acknowledge Saunders Consulting for the help in editing this manuscript. This research was undertaken in part using the MX2 beamline at the Australian Synchrotron, part of ANSTO, and made use of the Australian Cancer Research Foundation (ACRF) detector. Research reported in this publication was supported by NIH R01-DA038058 (AG), NIH R01-DA035263 (AG, HJGM), NIH F31-MH114316 (JIA), NIH P41-GM103712 (IB) and NIH P30-DA035778 (IB), and by the Intramural Research Program of the NIH, NIDA Z1A DA000606 (LS). The content is solely the responsibility of the authors and does not necessarily represent the official views of the National Institutes of Health.

# Additional information

## Competing interests

Amanda Duran: is now employed at Cyrus Biotechnology with granted stock options. However, all contributions to the present work were made during AD's graduate education at Vanderbilt University. The other authors declare that no competing interests exist.

## Funding

| Funder | Grant reference number | Author |
| --- | --- | --- |
| National Institute on Drug Abuse | R01-DA038058 | Aurelio Galli |
| National Institute on Drug Abuse | R01-DA035263 | Heinrich JG Matthies Aurelio Galli |
| National Institute of Mental Health | F31-MH114316 | Jenny I Aguilar |
| National Institute of General Medical Sciences | P41-GM103712 | Ivet Bahar |
| National Institute on Drug Abuse | P30-DA035778 | Ivet Bahar |
| National Institute on Drug Abuse | Z1A DA000606 | Lei Shi |

The funders had no role in study design, data collection and interpretation, or the decision to submit the work for publication.

## Author contributions

Jenny I Aguilar, Conceptualization, Formal analysis, Investigation, Visualization, Writing - original draft, Writing - review and editing; Mary Hongying Cheng, Josep Font, Alexandra C Schwartz, Kaitlyn Ledwitch, Formal analysis, Investigation, Visualization, Writing - original draft, Writing - review and editing; Amanda Duran, Samuel J Mabry, Andrea N Belovich, Yanqi Zhu, Investigation, Writing - review and editing; Angela M Carter, Formal analysis, Visualization, Writing - original draft, Writing - review and editing; Lei Shi, Funding acquisition, Investigation, Writing - review and editing; Manju A Kurian, Cristina Fenollar-Ferrer, Methodology, Writing - review and editing; Jens Meiler, Renae Monique Ryan, Ivet Bahar, Heinrich JG Matthies, Conceptualization, Supervision, Funding acquisition, Investigation, Writing - review and editing; Hassane S Mchaourab, Conceptualization, Formal analysis, Supervision, Funding acquisition, Visualization, Writing - original draft, Writing - review and editing; Aurelio Galli, Conceptualization, Supervision, Funding acquisition, Investigation, Writing - original draft, Writing - review and editing

## Author ORCIDs

Josep Font (iD) http://orcid.org/0000-0001-5284-1344
Alexandra C Schwartz (iD) https://orcid.org/0000-0002-7846-3772
Angela M Carter (iD) https://orcid.org/0000-0001-5766-3743
Lei Shi (iD) http://orcid.org/0000-0002-4137-096X
Ivet Bahar (iD) http://orcid.org/0000-0001-9959-4176
Heinrich JG Matthies (iD) https://orcid.org/0000-0003-3615-7571
Aurelio Galli (iD) https://orcid.org/0000-0001-7173-9345

## Decision letter and Author response
Decision letter https://doi.org/10.7554/eLife.68039.sa1
Author response https://doi.org/10.7554/eLife.68039.sa2

# Additional files

## Supplementary files
• Transparent reporting form

## Data availability

All relevant crystallography data are available from the corresponding author upon request. The maps and the coordinates of the refined models have been deposited into the Protein Data Bank. For LeuT wildtype PDB: 7LQJ, For LeuT R375A PDB: 7LQK and for LeuT R375D PDB: 7LQL.

The following datasets were generated:

| Author(s) | Year | Dataset title | Dataset URL | Database and Identifier |
|---|---|---|---|---|
| Font J, Aguilar J, Galli A, Ryan R | 2021 | Crystal structure of LeuT bound to L-Alanine | https://www.rcsb.org/7LQJ | RCSB Protein Data Bank, 7LQJ |
| Font J, Aguilar J, Galli A, Ryan R | 2021 | Crystal structure of the R375A mutant of LeuT | https://www.rcsb.org/7LQK | RCSB Protein Data Bank, 7LQK |
| Font J, Aguilar J, Galli A, Ryan R | 2021 | Crystal structure of the R375D mutant of LeuT | https://www.rcsb.org/7LQL | RCSB Protein Data Bank, 7LQL |

The following previously published datasets were used:

| Author(s) | Year | Dataset title | Dataset URL | Database and Identifier |
|---|---|---|---|---|
| Singh SK, Piscitelli | 2008 | Crystal structure of LeuT bound to | https://www.rcsb.org/ | RCSB Protein Data |

| | | | | |
|---|---|---|---|---|
| CL, Yamashita A, Gouaux E | | L-leucine (30 mM) and sodium | 3F3E | Bank, 3F3E |
| Gouaux E, Penmatsa A, Wang K | 2013 | X-ray structure of dopamine transporter elucidates antidepressant mechanism | https://www.rcsb.org/4M48 | RCSB Protein Data Bank, 4M48 |
| Yamashita A, Singh SK, Kawate T, Jin Y, Gouaux E | 2005 | Crystal structure of LEUTAA, a bacterial homolog of Na+/Cl–dependent neurotransmitter transporters | https://www.rcsb.org/2A65 | RCSB Protein Data Bank, 2A65 |
| Aravind P, Wang K, Gouaux E | 2015 | X-ray structure of Drosophila dopamine transporter bound to psychostimulant D-amphetamine | https://www.rcsb.org/4XP9 | RCSB Protein Data Bank, 4XP9 |

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
