## [Decision Letter]

**Acceptance summary:**

The authors present a comprehensive and technically sophisticated analysis of the structure, function and behavioral sequelae of a mutation in the human dopamine transporter. The analysis is translationally relevant since the mutation was identified in a patient suffering from Infantile Parkinsonism-dystonia, a rare but devastating condition that leads to early mortality. From a technical standpoint the paper is strong. The combination of behavioral and biochemical analysis in a transgenic animal with X-ray crystallography and modeling is unique and powerful. The partial pharmacologic rescue of the behavioral deficits further elevates this work and suggests that specific modulators of the transporter's structure that restore surface expression and function could be therapeutic for this disorder.

**Decision letter after peer review:**

Thank you for submitting your article "Psychomotor impairments and therapeutic implications revealed by a mutation linked with Infantile Parkinsonism-Dystonia" for consideration by *eLife*. Your article has been reviewed by 3 peer reviewers, including Rebecca Seal as the Reviewing Editor and Reviewer #1, and the evaluation has been overseen by Richard Aldrich as the Senior Editor.

Essential revisions:

1. The reviewers appreciate the use by the authors of different assays (cell culture and transgenic fly work) to examine the effects of a mutation on dopamine transporter cell surface expression and function as well as motor behavior, but the reviewers also take note of the different outcomes of these analyses with a gap existing between the loss of cell surface expression in cell culture and the down-regulation of TH in flies (what happens to DAT in flies is not experimentally addressed). Taken together with results from other studies on TH regulation, the authors suggest that the motor deficits in the flies result from a decrease in DAT cell surface expression (and/or transport of substrate) which causes an increase in extracellular DA levels (again not experimentally addressed), a subsequent decrease in TH levels and decrease in tissue/vesicular DA levels. This explanation seems plausible but also what happens in patients with respect to these findings is not experimentally addressed by the authors. The reviewers do not request new experiments, but request that the authors alter the text to discuss and clarify the aforementioned knowledge gaps within their study, including knowledge gaps with respect to the patient DA pathology. Also, the authors should make sure it is clear everywhere in the text if they are referring to results from cell culture or from the flies.

2. Additionally, DA regulates other CNS activities including wakefulness and these may contribute to the behavior of the fly mutant. The reviewers request that the authors discuss the possibility that the phenotype of the flies may not be strictly tied to deficits in motor activity and coordination. The authors should also discuss in the text the differences between their study and Ng et al. 2014 with respect to the cell culture results.

*Reviewer #1 (Recommendations for the authors):*

1. The CQ rescue of one of the behaviors seems to suggest that a reduction in surface levels of the transporter happens in flies, but the effect on motor behaviors was limited to flight initiation and did not include locomotion- was this because no change in locomotion was observed with the CQ? What does CQ do to locomotion in WT flies? It should be noted that it is difficult to ascertain from the western blot shown in Figure 7D, whether there was an increase in the mature form of the mutant hDAT with the lysosomal inhibitor CQ in cell culture. Importantly, neither TH levels nor surface DAT levels were examined in the flies with CQ. Ideally one would want the CQ experiment to be performed in a way that allows for the upregulation/restoration of TH in the flies (if this could indeed be a viable consequence of restoring surface DAT) as was done in cell culture (Figure 3) and motor behaviors including locomotion and flight, assayed.

2. As the authors suggest, the loss of TH^+^ neurons likely causes the decrease in DA that leads to the motor behaviors. It would be interesting to know whether there is a differential contribution of PPL1 DA neurons to different aspects of flight, locomotion, grooming that is reflected in the neurons that no longer release DA (were lost or no longer express TH).

Related to this, the authors proposed that a chronic increase in extracellular DA causes a down-regulation of TH in the flies as has been observed in other DAT mutants. It would thus be important to know whether there is indeed a loss of TH^+^ neurons in the patients with this mutation. The authors comment on this in the discussion but is postmortem tissue available?

The authors should state more clearly in the discussion which system was used for what experiment. The cell surface expression and transporter properties were measured in cell culture, but the motor deficits were assayed in the flies. It is important to avoid conflating these two systems. For example, the rescue in Figure 7 which tests effects of CQ on the transporter levels and function in cell culture and the motor effects in flies is an example where conflating the results could happen. What happens in cell culture may not be what is happening in the flies as discussed above.

*Reviewer #3 (Recommendations for the authors):*

The Reviewer has a few suggestions:

1. Discuss how a reduction in DA reuptake might affect tonic/burst firing.

2. It would help the reader if the Authors included a schematic diagram of the functional structure of DAT, indicating conformational changes that occur, and how R445C might affect DA transport, as nicely outlined in the Introduction.

3. A summary illustration of their hypothesis would be welcome.

4. P7, L159: Octopamine and TH should be referenced, or experiments should be performed to determine if epinephrine/norepinephrine are diminished in these flies

5. P9, L205: Parameters used for initiating and fast movements might be justified

6. P11, L249: Was an unbiased stereological approach used to measure the number of TH-positive neurons?

---

## [Author Response]

Essential revisions:1. The reviewers appreciate the use by the authors of different assays (cell culture and transgenic fly work) to examine the effects of a mutation on dopamine transporter cell surface expression and function as well as motor behavior, but the reviewers also take note of the different outcomes of these analyses with a gap existing between the loss of cell surface expression in cell culture and the down-regulation of TH in flies (what happens to DAT in flies is not experimentally addressed).

Now, we make clear within the text that in this study, DAT expression is exclusively determined in cell lines and not in isolated brain of *Drosophila*.

Taken together with results from other studies on TH regulation, the authors suggest that the motor deficits in the flies result from a decrease in DAT cell surface expression (and/or transport of substrate) which causes an increase in extracellular DA levels (again not experimentally addressed), a subsequent decrease in TH levels and decrease in tissue/vesicular DA levels. This explanation seems plausible but also what happens in patients with respect to these findings is not experimentally addressed by the authors. The reviewers do not request new experiments, but request that the authors alter the text to discuss and clarify the aforementioned knowledge gaps within their study, including knowledge gaps with respect to the patient DA pathology.

The reviewer is correct. In this study, we did not measure the possible increase in extracellular dopamine levels in fly brains. We now make this point clear within the text.

Also, the authors should make sure it is clear everywhere in the text if they are referring to results from cell culture or from the flies.

The text of the manuscript has been revised appropriately.

2. Additionally, DA regulates other CNS activities including wakefulness and these may contribute to the behavior of the fly mutant. The reviewers request that the authors discuss the possibility that the phenotype of the flies may not be strictly tied to deficits in motor activity and coordination.

This possibility is now discussed.

The authors should also discuss in the text the differences between their study and Ng et al. 2014 with respect to the cell culture results.

These differences have now been discussed.